# Arginine π-stacking drives binding to fibrils of the Alzheimer protein Tau

Luca Ferrari [1,2,6], Riccardo Stucchi[2,3,4,6], Katerina Konstantoulea [1,2], Gerarda van de Kamp [1,2], Renate Kos [1,2], Willie J.C. Geerts [2,5], Laura S. van Bezouwen[2,5], Friedrich G. Förster [2,5], Maarten Altelaar [2,4], Casper C. Hoogenraad [2,3] & Stefan G.D. Rüdiger [1,2]*

Aggregation of the Tau protein into fibrils defines progression of neurodegenerative diseases, including Alzheimer's Disease. The molecular basis for potentially toxic reactions of Tau aggregates is poorly understood. Here we show that π-stacking by Arginine side-chains drives protein binding to Tau fibrils. We mapped an aggregation-dependent interaction pattern of Tau. Fibrils recruit specifically aberrant interactors characterised by intrinsically disordered regions of atypical sequence features. Arginine residues are key to initiate these aberrant interactions. Crucial for scavenging is the guanidinium group of its side chain, not its charge, indicating a key role of π-stacking chemistry for driving aberrant fibril interactions. Remarkably, despite the non-hydrophobic interaction mode, the molecular chaperone Hsp90 can modulate aberrant fibril binding. Together, our data present a molecular mode of action for derailment of protein-protein interaction by neurotoxic fibrils.

[1] Cellular Protein Chemistry, Bijvoet Center for Biomolecular Research, Utrecht University, Padualaan 8, 3584 CH Utrecht, The Netherlands. [2] Science for Life, Utrecht University, Padualaan 8, 3584 CH Utrecht, The Netherlands. [3] Cell Biology, Neurobiology and Biophysics, Faculty of Science, Utrecht University, Padualaan 8, 3584 CH Utrecht, The Netherlands. [4] Biomolecular Mass Spectrometry and Proteomics, Bijvoet Center for Biomolecular Research and Utrecht Institute for Pharmaceutical Sciences, Utrecht University, Padualaan 8, 3584 CH Utrecht, The Netherlands. [5] Cryo Electron Microscopy, Bijvoet Center for Biomolecular Research, Utrecht University, Padualaan 8, 3584 CH Utrecht, The Netherlands. [6] These authors contributed equally: Luca Ferrari, Riccardo Stucchi. *email: s.g.d.rudiger@uu.nl

Protein aggregation is linked to a wide range of neurodegenerative disorders, including Alzheimer's, Parkinson's, and Huntington's diseases[1,2]. Protein fibrils are ubiquitous present in patients' brains affected by neurodegeneration[3]. For all neurodegenerative disorders, protein aggregation proceeds in a step-wise fashion, from structurally heterogenous oligomeric species to mature fibrils, the former considered to be the most toxic agents[4,5]. Remarkably, it is unclear why protein aggregates are toxic and how they react within the cellular environment of the neuron[6,7]. Intracellular aggregation or the protein Tau is hallmark of Alzheimer's disease and other fatal tauopathies[3,7,8]. Several factors play a role in the origin of Alzheimer's, such as extracellular amyloid formation of the Aβ peptide[9]. However, intracellular Tau aggregation is sufficient to induce neurodegeneration, correlates with cognitive impairment in humans and is necessary to mediate Aβ toxicity[6,10,11].

The mechanistic contribution to disease of Tau oligomers and fibrils remains largely elusive. Due to their structural heterogeneity and difficulties in isolation, it is difficult to point out which cellular processes they target[4,6]. However, non-physiological protein aggregates may result in new interactions within the cell, disturbing a variety of cellular processes and disrupting the protein quality control network, responsible for fibrils handling and disposal[12,13]. Major components of this network are two conserved energy-driven chaperone systems, Hsp70 and Hsp90[14]. They both participate in Tau clearance in physiological condition[15,16], however their contribution to neurodegeneration is still elusive. Hsp70 can disaggregate Tau fibrils, while Hsp90 buffers aggregation prone stretches of Tau[17–19]. Hsp90 decreased efficiency during aging may contribute to Tau aggregation, which in turn may dictate further collapse of chaperones activity[20].

Tau fibrils establish new abnormal interactions with either the insoluble proteome[21] or ER-associated protein complexes[22]. We recently showed that toxicity of an aggregation-prone variant of the protein Axin is caused by aberrant interactions established by oligomeric nano-aggregates formed in the cytoplasm[23]. Together, this made us wonder how Tau fibrils would interact with the soluble cytoplasmic component of the brain, where protein aggregation takes place. It is an open question whether common structural features govern the binding of interactors to Tau fibrils. It is of great interest to understand which interactions are engaged by Tau fibrils and their binding mechanisms, as this would reveal which cellular processes are targeted by Tau-dependent neurodegeneration and would offer novel therapeutic strategies.

We set out to understand at molecular level how aggregation of Tau modulates its interactions with proteins. Here we reveal that the key determinant for derailing Tau protein network is Arginine-driven π-stacking. This provides a molecular framework to understand aberrant interactions of neurotoxic fibrils. Interestingly, the avidity properties gained upon fibril formation attract a defined, new set of interactors, which contains disordered regions with a unique amino acidic footprint. Such regions are enriched in positively charged Arginines, and their substitution with positively charged Lysines impairs binding to Tau fibrils. This rules out charge-based interactions, pointing out the guanidinium group of the Arginine side chain as the determinant of binding to Tau fibrils via π-stacking. There is a cellular defence system suppressing Tau aggregation. We find that the Hsp90 chaperone stalls formation of Tau fibrils and reshapes their abnormal interactome. Thus, neurons are endowed with a powerful molecular machine that is able to counteract formation of Tau fibrils and their engagement with aberrant interactors.

## Results

**Separation of Tau monomers, oligomers, and fibrils.** To analyze potential interactome changes upon aggregation of Tau, it is crucial to precisely control fibril formation over time. We set out to biochemically characterize the aggregation process, from monomeric Tau at the start to mature fibrils as end-point. To this mean, we recombinantly produced a pro-aggregating ΔK280 variant of Tau Repeat Domain (Tau-Q244-E372, Tau-RD), with a FLAG tag to facilitate detection (FLAG-tagged Tau-RD-ΔK280, referred to as Tau-RD*)[24]. Tau Repeat Domain is an established Alzheimer model that aggregates more aggressively than the wildtype full length protein[25]. We induced its aggregation via heparin following an established procedure and collected samples as aggregation proceeded[26]. We then resolved these samples on density gradients spread over 12 fractions and detected them with a fluorescent antibody (Fig. 1a).

At time 0, Tau-RD* sedimented on top of the gradient (Fractions 1 and 2, tube I), consistent with its expected monomeric nature at the start of the experiment. After 1 h (tube II), Tau-RD* sedimented further down until fraction 3, indicating the appearance of oligomeric, nanometer-scale aggregates (nano-aggregates). At 8 h (tube III), Tau-RD* spreaded until fractions 7, consistent with the growth of increasingly large aggregates. Finally, after 24 h (tube IV), we detected no Tau-RD* in fraction 1 anymore, indicating that all Tau-RD* aggregated into either oligomers or

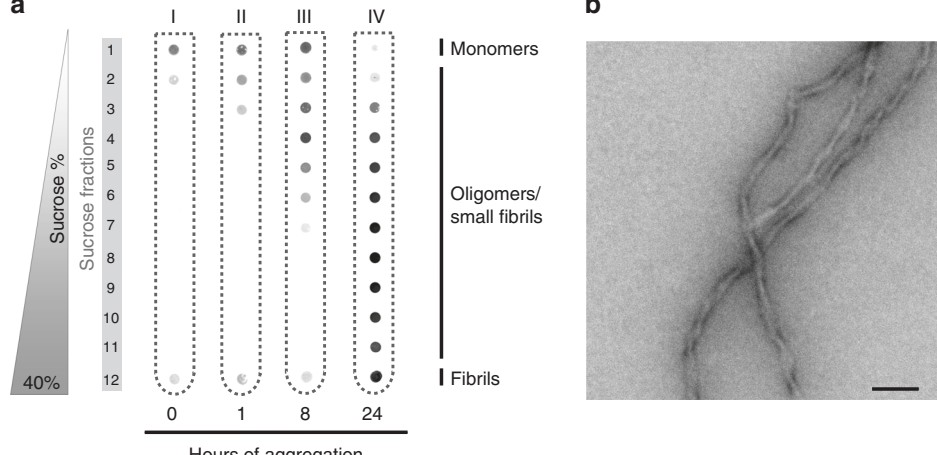

**Fig. 1 Precise control of Tau aggregation. a** Time course of Tau-RD* aggregation monitored by density gradients. Samples were centrifuged for 2.5 h at 200,000×*g*. **b** Transmission electron microscopy of Tau-RD* fibrils, timepoint 24 h. Scalebar 100 nm.

fibrils, reflected by the presence of aggregates from fraction 2 to 12. Transmission electron microscopy (TEM) characterized Tau-RD* fibrils at 24 h as negatively stained paired helical filaments with periodicity of 50–100 nm, consistent with previous observations[27] (Fig. 1b and Supplementary Fig. 1). Thus, our procedure generates the full spectrum of Tau species for aggregation-dependent interactome analysis, representing monomers, oligomeric nano-aggregates, and mature fibrils.

**Tau fibrils target specific protein families**. Next, we wondered how Tau reacts at different aggregation stages with the soluble neuronal proteome. We exposed the FLAG-tagged Tau-RD* at different stages of aggregation during a 24 h period with rat brain lysates depleted of their insoluble components (Fig. 2a). The appearance of Tau aggregates changed during aggregation. While oligomers were too small to be visible in negative stain EM images after 90 min, we observed fibrils after 4.5 h (Supplementary Fig. 1). Over 24 h, we pulled down potential interactors of aggregating Tau with an anti-FLAG antibody and revealed their identity via mass spectrometry in meaningful intervals. For each protein, we determined the abundance by counting either the peptide spectrum matches (PSMs) or intensities of the peptide peaks at each aggregation stage. We compared the protein spectrum at each timepoint ($t_x$) to the one of monomeric Tau-RD* without heparin ($t_{0-}$) and plotted changes as logarithmic function. This setup allowed us to monitor proteome changes (Supplementary Data 1 and 2).

We uncovered striking interactome changes as Tau-RD* aggregation proceeds (Fig. 2b). A sub-group of interactors showed decreased levels compared to the monomeric reference values (Fig. 2b, purple box $t_{24}$, blue interactors). Another sub-group of interactors showed binding levels similar for the fibril fractions as for the monomers. Remarkably, however, they bound stronger to early-stage Tau nano-aggregates (Fig. 2b, purple box $t_1$). Finally, a sub-group of the proteome increasingly associated with Tau upon progression of fibril formation, reaching its maximum as mature fibrils appeared (Fig. 2b, purple box $t_{24}$, red interactors). The time-dependent change in protein–protein interactions implies that fibrils and oligomers differ in binding properties to other proteins. Thus, progressive aggregation gradually rewires Tau interactome, with different aggregation species interacting with different molecular partners.

We wondered whether the aberrant interactors of Tau aggregates share common functional properties by classifying them via Gene Onthology enrichment analysis, a tool to cluster proteins with similar biological functions (GO Consortium, 2017[28]). We first focused on monomeric-specific interactor lost upon aggregation, with only 25% of interactors belonging to functional clusters, namely protein phosphorylation regulators (16%) or RNA-binding proteins (9%) (Fig. 2c). Notably, members of the COPI complex are the most prominent interactor of nano-aggregates ($t_1$). They neither bind to monomers nor fibrils, indicating that this complex prefers binding to Tau oligomeric species (Fig. 2c, dashed purple box).

When analyzing the interactome of the Tau-RD* fibrils, remarkably 66% of the newly attracted interactors belong to only three major functional clusters (Fig. 2d), namely RNA-binding proteins (40% of the interactors), regulators of protein phosphorylation (20%), microtubule-associated proteins (MAPs, 6%), with the rest of the proteins showing no GO enrichment (various proteins, 34%). We confirmed these results in three biological replicates for both monomers and fibrillar interactors (Supplementary Figs. 2 and 3). We analyzed these data sets by two different approaches, by PSM counting and by quantifying area-under-the-curve of extracted ion chromatograms (Supplementary

Figs. 4 and 5). Together, these analyses revealed that our findings are reproducible and independent of any potential methodological bias. Thus, we conclude that Tau aggregates attract proteins connected to a very limited set of biological processes, exchanging proteins connected to the same functional networks and increasing their relative abundance, whereas the rest of the cellular activity seems to not be disturbed by Tau fibrils. Strikingly, COPI preferentially interact with early stage aggregates and MAPs with late stage fibrils, suggesting that different types of aggregates may differ in their gain of function aberrant phenotypes.

**Arginine side chains mediate fibril interactions**. We hypothesized that shared biological activity may underline common sequence features. We noted that RNA and microtubule-binding proteins and regulatory proteins often contain intrinsically disordered regions. Therefore, we determined the degree of disorder for each fibril-binding protein using the disorder predictor MetaDisorder, particularly accurate as it is the consensus of 15 primary algorithms[29]. We then plotted the disorder content per functional cluster. All functional clusters, even non-clustered fibrils binders, showed high disorder content, with medians at least 15% higher in absolute disorder than the average disordered percentage in the human proteome[29] (Fig. 3a). Most dramatic is the MAP cluster, which is 95% disordered. We conclude that Tau fibrils attract proteins enriched in intrinsically disordered regions. It is unusual for intrinsically disordered proteins to directly interact with each other[30], therefore we can also conclude that fibrillation endows Tau with new properties to engage in protein–protein interactions.

We wondered whether disordered stretches captured by Tau fibrils showed specific sequence properties. To reveal such a bias, we compared the frequency of each amino acid in the disordered regions of fibril interactors to the average frequency in the whole disordered proteome[31]. Remarkably, our analysis showed that chemical composition of disordered regions of fibril-specific interactors dramatically differed from that of typical disordered regions (Fig. 3b). In particular, prolines and valines were significantly depleted, although they are the most abundant components in the disordered proteome. Conversely, arginines and methionines stood out as the most significantly enriched amino acids ($p < 0.0001$, Wilcoxon paired non-parametric $t$-test). Arginines had higher frequency when compared to methionines (5% of total residues in disordered regions for arginine, 1.6% for methionine, Fig. 3b). More than 85% of the interactors had at least one long disordered region (30+ residues), with 90% of these regions containing at least two Arginines. Thus, there is a predominant role for Arginine in intrinsically disordered regions binding to Tau fibrils.

When comparing the disordered segments in fibrils binders to the folded proteome, arginine and methionine were enriched while most of the amino acids were depleted (Fig. 3c). Also here, arginine stood out again as the most significantly enriched amino acid ($p < 0.0001$, Wilcoxon paired non-parametric $t$-test). These differences in footprint indicate a unique signature of proteins binding to Tau fibrils. In particular the bias for Arginine-rich disordered regions differs from both typical disordered segments and globular proteins.

**Arginine π-stacking crucial for binding to Tau fibrils**. Next, we aimed to reveal the molecular mechanism of interaction of Tau fibrils with intrinsically disordered regions. Tau-RD has a positive net charge at physiological pH ($pI = 9.7$ for Tau-RD, 9.3 for Tau-RD*), making us wonder why positively charged Arginines would be so enriched in Tau aggregation-specific interactors. Notably,

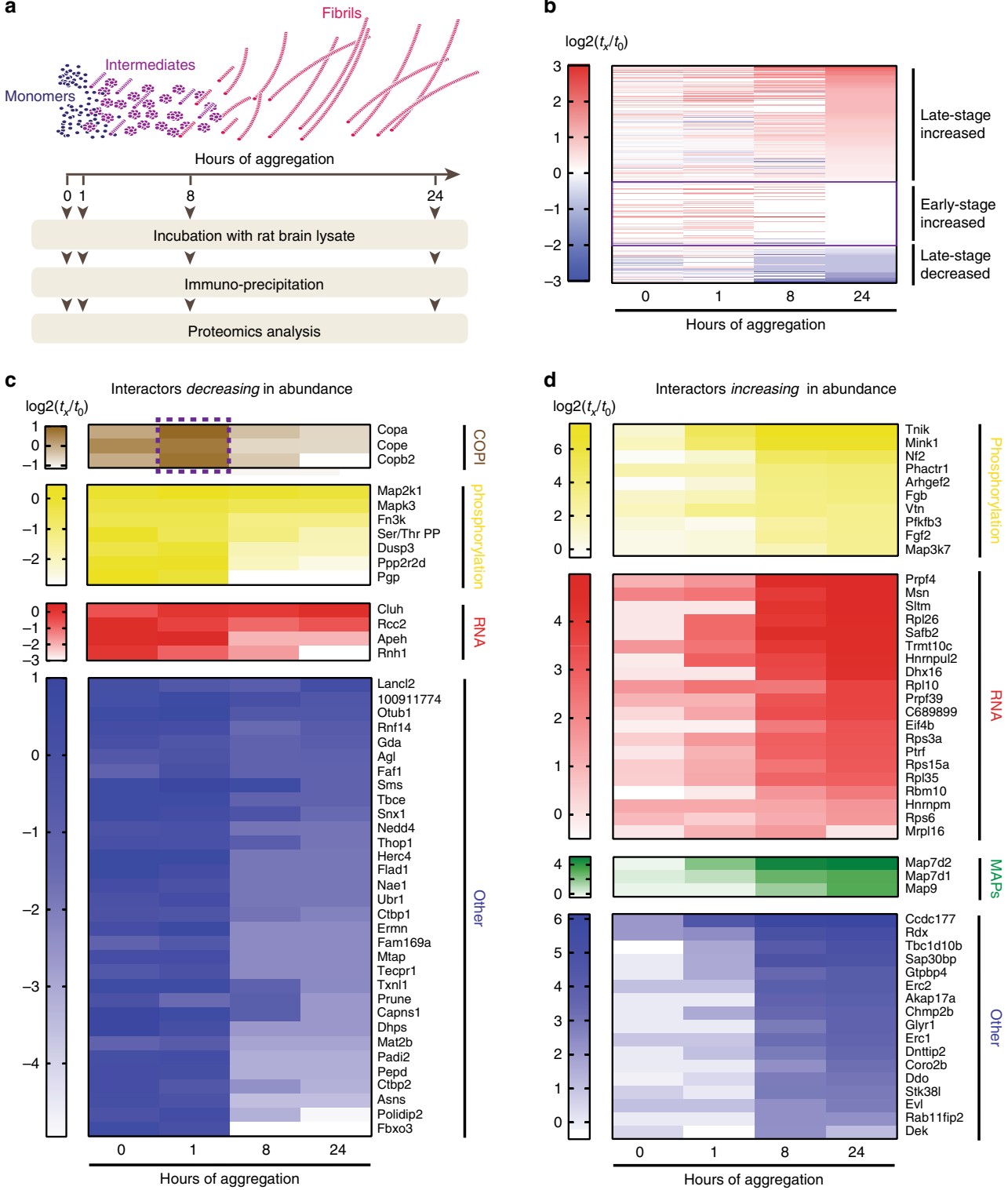

**Fig. 2 Tau interactome rewires during aggregation. a** Experimental setup to highlight interactome changes upon Tau aggregation. **b** Heat map of MS-identified proteins of Tau-RD* aggregates (0, 1, 8, and 24 h of aggregation). Heatmap shows relative enrichment of different time points ($t_x$) compared to monomeric Tau-RD* without heparin ($t_{0\_}$). Interactors are sorted by trend (late-stage decreased, early-stage increased, late-stage increased). **c** Heat map showing an unbiased selection of Tau-RD* lost proteins (decreasing in abundance) upon aggregation, sorted by colored functional clusters (GO-term analysis). Dashed purple box highlights interactors increasing in abundance at 1 h, then decreasing as aggregation proceeds. See also Supplementary Data 1. **d** Heat map showing an unbiased selection of Tau-RD* sequestered proteins (increasing in abundance) upon fibril formation, sorted as for **c**. See also Supplementary Data 1.

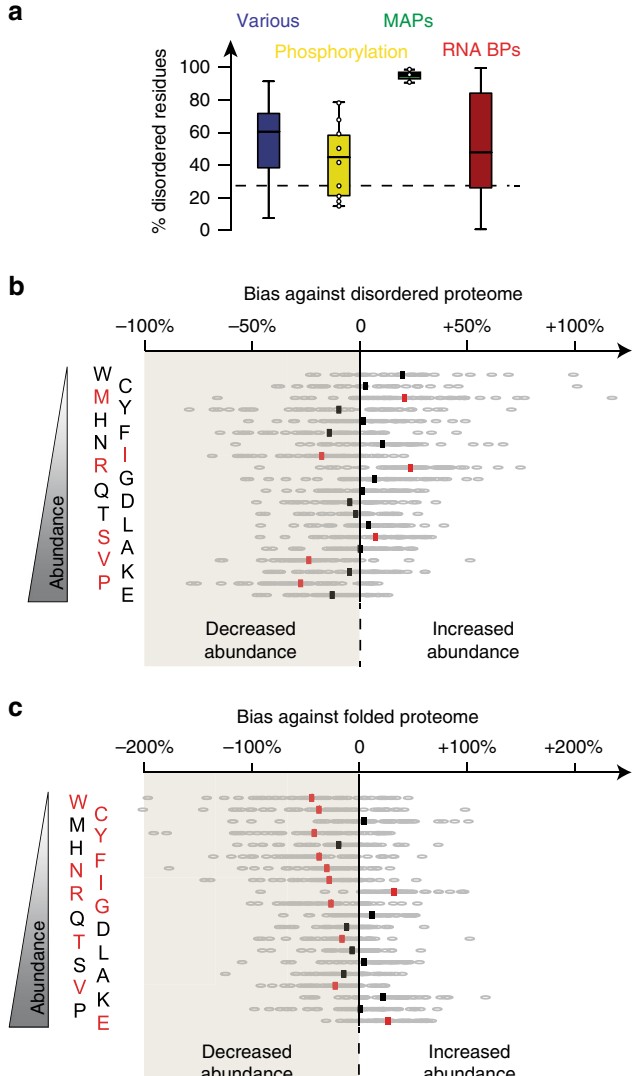

**Fig. 3 Tau aggregation-specific interactors are enriched in disordered sequences. a** Prediction of percentage of disordered residues of Tau-RD* fibril-specific interactors (timepoint 24 h) using MetaDisorder[29], sorted by functional clusters (same color-code of Fig. 2) and represented as box plot (box comprises first to third quartiles, whiskers indicate maximum and minimum values). Dashed line indicates the average disorder percentage for human proteome[68]. Dots indicate individual proteins when cluster has $n < 10$. **b** Amino-acidic footprint of disordered regions of Tau-RD* fibril-specific interactors, expressed in fold-changes against the abundance of each amino acid in the disordered regions of the whole human proteome[68] (red, $p \leq 0.0001$). Amino acids on the y-axis are sorted by abundance in the disordered proteome. **c** As in **b**, but against folded regions of the whole human proteome[31] (red, $p \leq 0.0001$). Amino acids on the y-axis are sorted by abundance in the disordered proteome, for comparison with **b**.

next to charge–charge interactions, Arginine can also engage in protein–protein interactions via the delocalized π-system of $sp^2$-hybridized atoms of its guanidinium group, stacking on top of another delocalized π-system[32]. This π-stacking is particularly known for aromatic rings, e.g. by stabilizing the DNA double helix. In line with this, we noted that there is no significant bias for the other positively charged residue, lysine (Fig. 3b), whose side chains contain only $sp^3$-hybridized atoms and are thus unable to engage in π–π interactions, suggesting that π-orbitals and not charge could be the determinant for binding to fibrils. We set out to test this hypothesis by analyzing the impact of

arginine to lysine exchanges in a fibril-binding protein, an established test to verify the contribution of π–π interactions[32].

We chose the N-terminal domain of the fibril-specific protein Map7 for this test (Fig. 4a). This domain has extended disordered stretches enriched in Arginines (8.4% of disordered residues), next to a folded coiled coil domain, making it a valuable candidate to test arginine-driven binding to fibrils. We designed and purified two HA-tagged truncations of Map7-M1-S227, wildtype and an $R_{12}K$ variant, where we replaced all arginines in the disordered regions of the protein with lysines. The $R_{12}K$ substitution had no effect on the net charge of the protein (Fig. 4a). The circular dichroism spectra of both proteins were identical and they both indicated the presence of α-helixes and disordered stretches (Fig. 4c). Thus, exchanging arginines to lysines in the disordered segments of the N-terminal fragment of Map7 does neither alter its charge nor its secondary structure.

Next, we set out to test whether the exchange of Arginines to Lysines would affect binding to fibrils. To this end, we first generated Tau-RD* fibrils and then tested their chemical binding to Map7 truncations. We resolved protein complexes via density gradients, using different antibodies to detect Tau-RD* and Map7. Map7 wt and $R_{12}K$ sedimented throughout the tube (Fig. 4d, anti-HA blot, Tube I and III), whereas Tau-RD* fibrils sedimented to the bottom half of the tube (Fig. 4d, anti-FLAG blot, tube II and IV, fractions 7–12). The homogenous spread of Map7 alone throughout the tube could be explained by the tendency of disordered regions to show higher apparent molecular mass compared to order structures[33]. When Map7 wt and Tau-RD* fibrils reacted together, however, the entire Map7 population sedimented in the heaviest fraction, indicating its association with the largest fibrillar structures (Fig. 4d, anti-HA blot, tube II, fraction 12), consistent with the proteomics analysis (Fig. 2c). Conversely, when Map7 $R_{12}K$ and Tau-RD* fibrils reacted, only a part of Map7 population sedimented in the heaviest fraction, whereas another part distributed in the lightest fractions, indicating only partial association with largest fibrillar structures (Fig. 4d, anti-HA blot, tube IV, fractions 1–5 and 12; quantification in Supplementary Fig. 6). Since both constructs have same charge and structural properties but differ in their ability to engage in π–π interactions, we conclude that π–π interactions are key forces governing binding to fibrils.

**Hsp90 stalls Tau aggregation.** Aberrant interactome rewiring is linked to cellular toxicity[23]. It would be therapeutically valuable to find endogenous proteins able to modulate such aberrant transition in interactome. We looked for such endogenous player in the protein quality control network, a pool of proteins that ensures the health of the proteome by removal of misfolded and aggregated proteins[1]. A key player in this network is the molecular chaperone Hsp90, which also controls Tau levels in the cell. Hsp90 forms a complex with Tau buffering its aggregation-prone regions[18] and controls the degradation of Tau monomers[34].

To understand the effect of Hsp90 on Tau aggregation dynamics, we wondered how this molecular chaperone would interfere with the formation of Tau fibrils. To monitor aggregation of Tau-RD* in the presence of increasing concentrations of Hsp90, we induced aggregation in the presence of Thioflavin T, a dye that recognizes amyloid structures (Fig. 5a and Supplementary Fig. 7). After aggregation reached plateau for all conditions, we resolved aggregating samples on density gradients (Fig. 5b). In a dose-dependent manner, Hsp90 suppressed Tau-RD* to form higher molecular weight species, decreased amyloid content and prevented fibril formation altogether. Hsp90 also altered morphology of the aggregates: instead of fibrils, only smaller structures were visible in the TEM

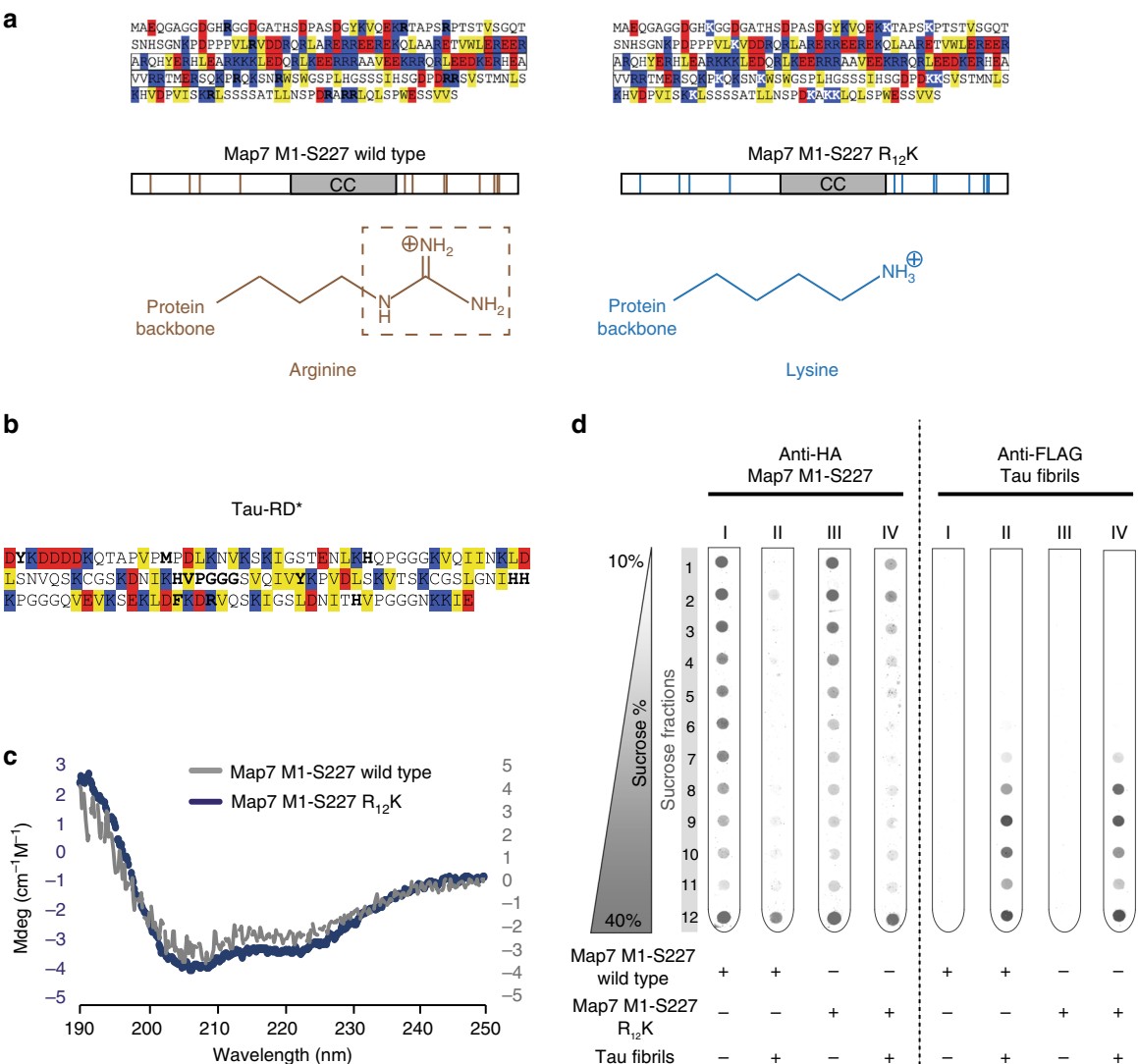

**Fig. 4 π-stacking of arginines drive binding to Tau fibrils. a** Graphic scheme of the two Map7 truncations used in this study. Brown and blue bars, respectively, indicate arginines and lysines in the disordered regions of the two truncations. Formulas of the two amino acids are presented, with the arginine guanidinium group (capable of π-stacking) boxed. CC = coiled-coil domain, as indicated by UniProt (access code: O88735). Sequences of Map7 M1-S227 truncations colored by hydrophobicity (yellow boxes), negative charges (red boxes), and positive charges (blue boxes) are reported on top of the cartoons; white Ks indicate arginines replaced with lysines. Boxed letters comprise the coiled-coil domain. **b** Tau-RD* sequence, colored as in **a**. Bold letters represent potential π-stacking sites, including the VPGGG motif[32,50]. **c** Circular dichroism of Map7 M1-S227 truncations. **d** Binding of Map7 1M-227S truncations to Tau-RD* fibrils, assessed by density gradients. Tubes (I–IV) were either blotted against HA tag or against FLAG tag, targeting respectively Map7 truncations or Tau-RD* fibrils. Samples were centrifuged for 2.5 h at 200,000×g.

images when aggregation was performed in the presence of Hsp90 (Fig. 5c). Thus, Hsp90 has a dramatic effect on Tau aggregation dynamics, derailing its aggregation toward non-fibrillar nano-aggregates with decreased amyloid content.

**Hsp90 modulates fibril interactome**. Next, we wondered whether aggregates with different structural features should attract different binding partners. To address this question, we monitored Tau-RD* aggregation in the presence or absence of Hsp90 and described the changes in abundance of pulled-down inter-actors in these two conditions (Fig. 6a, Supplementary Data 3 and 4). Stunningly, none of the fibril-specific interactors were enriched, whereas many of them decreased in abundance for all functional clusters when Hsp90 modulated the aggregation (Fig. 6b and Supplementary Fig. 8). The only interactor that showed a significant enrichment was α-synuclein (SNCA), a

well-known Hsp90 partner, suggesting that Hsp90 may co-aggregate with nano-aggregates. As expected, a subgroup of interactors, associated to Tau monomer but not to Tau fibrils (FBXO3, CTBP1, CTBP2, and POLDIP2, Supplementary Fig. 2) did not change in the presence of Hsp90. Also, the Hsp90-stalled nano-aggregates did not associate with COPI, in contrast to the ones formed in its absence. Taken together, these results suggest that Hsp90 drastically reshapes Tau aggregation-specific inter-actome, blocking the interactions associated to pathological Tau fibrils and molding the interactome toward a more physiological state.

## Discussion

We tracked interactome changes associated to the formation of Tau fibrils, a hallmark of Alzheimer's disease. Tau fibrils results from the aggregation of monomeric Tau, known to stabilize

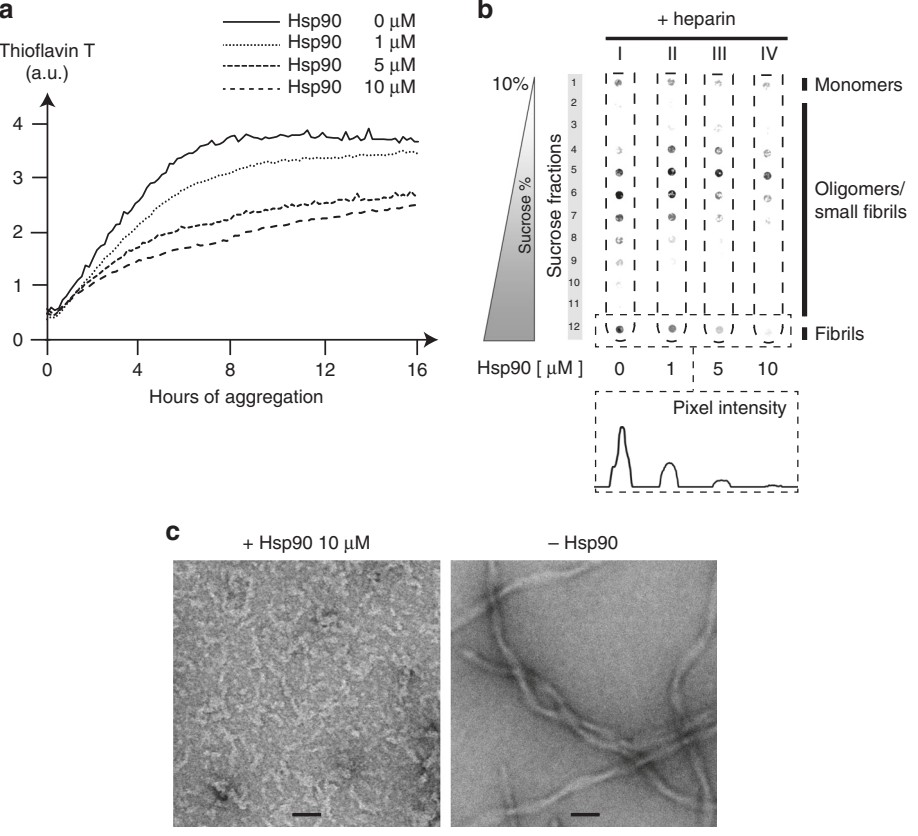

**Fig. 5 Hsp90 derails Tau aggregation toward non-fibrillar nano-aggregates. a** Thioflavin T assay to detect decrease of amyloid content in the presence of increasing concentrations of Hsp90 (0, 1, 5, 10 μM). Fluorescence is expressed as arbitrary unit (a.u.). **b** Density gradients to detect inhibition of fibril formation in the presence of increasing concentrations of Hsp90 (0, 1, 5, 10 μM). Samples from **a** at 16 h were loaded directly onto density gradients and resolved by centrifugation for 2.5 h at 200,000×*g*. Inset shows quantification of the fibril-containing fraction for each sample. **c** Transmission electron microscopy of Tau-RD* aggregating samples in the presence or absence of 10 μM Hsp90 after 2 days of aggregation. Scale bar 50 nm.

microtubules in neurons. As Tau fibrils form, they attract abnormal interactors endowed with long disordered stretches with specific amino acidic bias. Arginine bias plays a crucial role, as arginine-to-lysine substitutions partially impairs the binding to Tau fibrils. Our findings indicate arginine π-stacking as the chemical force governing Tau fibrils attraction potential.

Our experimental setup focuses on the soluble fraction of the cytoplasm, the environment where fibrils naturally accumulate during the progression of Alzheimer[6]. Proteomics studies concerning Tau fibrils reveal interactomes of insoluble membrane-enriched fractions co-pelleted with Tau fibrils and their ER-associated components[21,22]. Here we provide data sets revealing interactions of Tau fibrils with the soluble components of the brain, to which the protein is exposed when aggregating in disease. We focused on the interactions established by Tau-RD, as this stretch harbors the amyloid toxic fold[35,36]. Therefore, the interactome we obtained gives a comprehensive picture of amyloid-associated binders. Our approach may have missed potential interactors of the less hydrophobic N- and C-terminal stretches missing in Tau-RD. However, as aggregation of the Tau-RD fragment causes memory loss in vivo[25], it is likely that interactors of the amyloidogenic core region are potentially most relevant for neurodegeneration.

Our setup accelerates the process of Tau aggregation from years in vivo to hours in vitro and highlights proteomic re-arrangements over time, adding a time dimension to proteomics studies. Thus, our study mimics the temporal dynamics of Tau aggregation spanning several decades from initial seeds to mature fibrils, with different aggregation stages having different reactivity with the cellular environment (Fig. 2b). Scattered

experimental evidences support the targeting of RNA, microtubule, and phosphorylation dynamics by Tau fibrils[37–40]. Our work shows how these processes are coherently linked by interactome re-wiring of Tau from its monomeric, physiological protomer to its polymeric, toxic aggregate.

We fractionated Tau aggregates over time to overcome the problem that the dynamic nature of the aggregation process would preclude us from obtaining a defined, mono-disperse Tau nano-aggregate[6]. We show that early-stage aggregates specifically attract COPI components. These interactors decrease in abundance as fibrils form, highlighting the exchange of interactors as aggregation proceeds and the specific reactivity of Tau oligomers. Our data can explain Golgi fragmentation observed in neurons of Alzheimer brains without tangles[41]. They also suggest a route for Tau oligomers to interact with the vesicular transport system, which may support release outside neurons and some prion-like activities[42].

We find that Tau fibrils hijack protein belonging to three cellular processes: (1) RNA-related processes, (2) cytoskeletal dynamics via MAPs, and (3) phosphorylation equilibria (Fig. 2c). Regarding RNA-related processes, the interactome of monomeric Tau is already highly enriched in components of the ribonucleoproteome[37], and Tau fibrils have been shown to impair RNA translation[38]. Tau fibrils target also MAPs, linked to cytoskeletal dynamics. MAPs have been shown to act together to regulate microtubule dynamics and cargo transport, with Map7–Tau imbalances disrupting axonal transport and neuronal growth[39]. Such imbalances in axonal transport are a common marker of Tau-related neurodegenerative disorders[40].

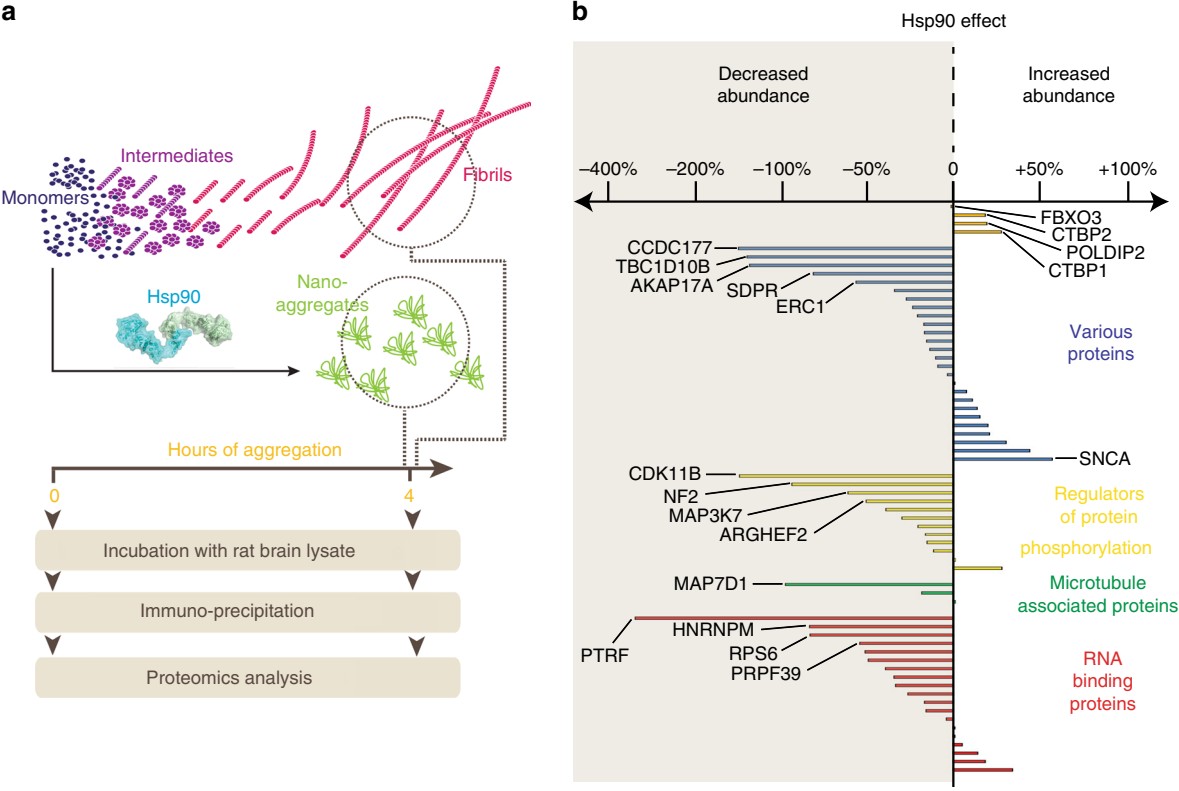

**Fig. 6 Hsp90 remodels Tau aggregation-specific interactome. a** Experimental setup to compare interactome changes in the presence or absence of Hsp90. **b** Bar graph showing Hsp90-dependent modulation of Tau-RD* aggregation reflected as differences in enrichments for specific proteins sequestered by Tau-RD* fibrils in the presence/absence of Hsp90. Proteins are clustered in colored functional clusters as for Fig. 2c and d. See also Supplementary Data 3.

The fact that Tau fibrils attract kinases and other phosphorylation regulators relates to hyperphosphorylation of Tau in Alzheimer[6,7]. Our data show that Tau fibrils target both kinases and protein phosphatases, two classes of proteins crucial for neuronal survival during neurodegeneration[43,44]. Interaction of Tau fibrils with these proteins may contribute to disrupt pro-survival pathways. While phosphorylation speeds up fibril formation of Tau, remarkably Tau fibrils isolated from patients' brain do not show regular phosphorylation patterns that would be visible in the structure[36,45]. Conversely, recent findings revealed how Tau hyperphosphorylation induces phase separation and consequent assembly of Tau fibrils[46]. However, the mechanistic impact of phosphorylation on Tau aggregation is yet to be understood.

The fibril-specific interactome of Tau is highly enriched in disordered stretches, compared to average disorder propensity of the human proteome. This is remarkable given that Tau itself is an intrinsically disordered protein. Intriguingly, the abundance of disordered stretches is a characteristic of interactors binding to Huntingtin fibrils[47] or to artificial proteins with propensity to form amyloid fibrils[48], suggesting that the preference for highly disordered binders is a general feature of amyloids.

We found a striking enrichment of arginine and methionine residues in disordered regions of Tau fibril binders, while prolines and the hydrophobic residues isoleucine, valine, and phenylalanine were significantly depleted, highlighting a unique footprint of Tau fibrils binders. The most prominent role is assigned to arginine. Notably, also the rarer methionine can establish π–π interactions and is enriched in fibril interactors (Fig. 3)[49]. In contrast, aromatic amino acids, which can also engage in π-interactions were not enriched, possibly because they are less well tolerated in disordered stretches.

Remarkably, the other positively charged side chain, lysine, does not contribute to binding, despite their positive charge shared with arginines, pointing to the role of arginine as π-stacking[32]. Arginine is better suited than lysine to form multiple hydrogen bonds, especially to phosphates and carboxyl groups. However, negatively charged side chains are depleted in the positively charged Tau-RD, excluding a major role for Coulomb interaction between Tau fibrils and arginine-rich interactors (Fig. 4b). Thus, this points at a key role for the π-stacking potential of the arginine side chain to drive aberrant interactions to Tau fibrils.

We mapped potential π-stacking residues in the Tau sequence, revealing 10 potential sites, including a VPGGG motif, which is known to engage in π-stacking in other context[32,50] (Fig. 4b, bold letters). Tau fibrillization stacks these sites, possibly increasing the avidity of arginine-rich binders and favoring their binding to fibrils over monomers. Thus, our work describes interactome re-arrangements as a key molecular component driving interactome re-arrangements, namely arginine side chains capable of engaging in π–π interactions.

Arginine π-stacking in disordered regions is also a fundamental force driving liquid–liquid phase separation, a process that forms membrane-less compartmentalization within a cell[51,52]. The number of arginine residues is a key factor for phase separation propensity[53]. So far, the role of arginine-driven π–π stacking was shown only for physiologically relevant proteins. We turned the table, using non-physiological aggregates to show that their interactions are based on the same chemical principles driving phase separation. Our paradigm explains recent phenomenological data, showing co-influence of aggregation dynamics between Tau and phase-separating proteins[54,55].

Interestingly, Tau fibril-specific interactome contains proteins from the heteronuclear ribonucleoparticle family (Fig. 2a and

Supplementary Fig. 3), whose protein architecture promotes phase separation[56]. Thus, Tau fibrils can attract phase-separating proteins, potentially increasing their local concentration and promoting their co-aggregation. Notably, also Tau and Tau-RD can phase-separate before forming mature, downstream fibrils[46,57]. Fibrillation and phase separation may be mechanistically related at molecular level as both mutations and post-translational modifications that prevent fibrillation can also prevent phase separation[46,58]. Such Tau condensates may also induce aberrant condensation of partner proteins[59], which could be an intriguing factor modulating the interactome. However, a significant contribution of Tau droplets to the binding of phase-separating proteins observed in our study is unlikely because we generated aggregates at physiological ionic strength, which precludes phase-separation of Tau[57,60]. It did not escape our attention, though, that both phase separation and interactome modulation of fibrils both rely on π–π stacking. Taken together, our data mechanistically support a tight crosstalk between physiological and pathological aggregates. In terms of fundamental chemical contacts, physiological and pathological aggregates are two sides of the same coin.

Interestingly, arginine is enriched in several chaperone-binding motifs, including Hsp70, J-domain co-chaperones, SecB, and trigger factor[61–64]. We speculate that arginine may have specific but not yet understood implications for protein homeostasis and aggregation. In this context it is interesting that the abundant cytosolic Hsp90 chaperone[65] may have an upstream effect in avoiding the engagement of Tau fibrils with their abnormal interactors by buffering these exposed stretches in the cytoplasm. Hsp90 may therefore act on two levels on Alzheimer-related Tau fibrils, first by controlling Tau homeostasis and second by preventing fibrils binding to their abnormal interactors. Understanding how the manipulation of Hsp90 affect the biology of Tau fibrils may provide a tool to target derailment of protein quality control in Alzheimer's disease.

## Methods

**Purification of Tau-RD\***. We overproduced N-terminally FLAG-tagged (DYKDDDDK) human Tau-RD (Q244-E372, also referred to as K18, with pro-aggregation mutation ΔK280) recombinantly in *E. coli* BL21 Rosetta 2 (Novagen), with additional removable N-terminal His$_6$-Smt-tag (His$_6$-Smt-tag amino acidic sequence: MGHHHHHHGSDSEVNQEAKPEVKPEVKPETHINLKVSDGSSEIFFK IKKTTPLRRLMEAFAKRQGKEMDSLRFLYDGIRIQADQTPEDLDMEDNDII EAHREQIGG; see Supplementary Table 1 for information about primer sequences). Cells were harvested, flash-frozen in liquid nitrogen and stored at −80 °C until further usage. Pellets were thawed in a water bath at 37 °C and resuspended in 50 mM HEPES–KOH pH 8.5 (Sigma-Aldrich), 50 mM KCl (Sigma-Aldrich), 1/2 tablet/50 ml EDTA-free protease inhibitor (Roche), 5 mM β-mercaptoethanol (Sigma-Aldrich). Cells were disrupted by an EmulsiFlex-C5 cell disruptor (Avestin). Lysate was cleared by centrifugation, filtered with a 0.22 μm polypropylene filter (VWR) and supernatant was purified using an ÄKTA purifier chromatography system (GE Healthcare). First, protein was loaded onto a POROS 20MC (Thermo Fischer Scientific) affinity purification column in 50 mM HEPES–KOH pH 8.5, 50 mM KCl, 5 mM β-mercaptoethanol, eluted with a 0–100% linear gradient (5 column volumes, CV) of 1 M imidazole. Fractions of interest were collected and concentrated in a buffer concentrator (Vivaspin, cut-off 10 kDa) to final volume of 3 ml. The concentrated sample was desalted with a PD-10 desalting column (GH Healthcare) in 50 mM HEPES–KOH pH 8.5, 1/2 tablet/50 ml Complete protease inhibitor (Roche) and 5 mM β-mercaptoethanol. The His$_6$-Smt tag was removed by Ulp1 treatment, shaking at 4 °C over night. Next day, protein was loaded onto a POROS 20HS (Thermo Fischer Scientific) cation exchange column equilibrated with 50 mM HEPES–KOH pH 8.5. Protein was eluted with a 0–100% linear gradient (15 CV) of 2 M KCl (Carl Roth). Fractions of interest were collected and loaded onto a HiLoad 26/60 Superdex 200 pg (GE Healthcare Life Sciences) size exclusion column equilibrated with aggregation buffer (25 mM HEPES–KOH pH 7.5, complete protease inhibitors (1/2 tablet/50 ml), 75 mM KCl, 75 mM NaCl, and 10 mM DTT). Fractions of interest were further concentrated to the desired final concentration using a concentrator (Vivaspin, cut-off 5 kDa). Protein concentration was measured using an ND-1000 UV/Vis spectrophotometer (Nanodrop Technologies) and purity was assessed with SDS–PAGE. Protein was aliquoted and stored at −80 °C.

**Purification of MAP7 truncations**. We overproduced N-terminally HA-tagged (YPYDVPDYA) mouse Map7 truncations (M1-S227, both wild type and R$_{12}$K), recombinantly in *E. coli* BL21 Rosetta 2 (Novagen), with additional removable N-terminal His$_6$-Smt-tag. Cells were harvested, flash-frozen in liquid nitrogen and stored at −80 °C until further usage. Pellets were thawed in a water bath at 37 °C and resuspended in 50 mM phosphate buffer pH 8 (Sigma-Aldrich), 150 mM KCl (Sigma-Aldrich), 1/2 tablet/50 ml EDTA-free protease inhibitor (Roche), 5 mM β-mercaptoethanol (Sigma-Aldrich). Cells were disrupted by an EmulsiFlex-C5 cell disruptor (Avestin). Lysate was cleared by centrifugation, filtered with a 0.22 μm polypropylene filter (VWR) and supernatant was purified using an ÄKTA purifier chromatography system (GE Healthcare). First, protein was loaded onto a column with Ni-IDA resin (GE Healthcare). Proteins were eluted by 250 mM imidazole (Sigma-aldrich) in 50 mM phosphate buffer pH 8.0, 150 mM NaCl (Sigma-Aldrich), complete protease inhibitors (1 tablet/100 ml) (Roche) and 5 mM β-mercaptoethanol (Sigma-Aldrich).The His$_6$-Smt tag was removed by Ulp1 treatment at 4 °C over night, while dialyzed against 50 mM phosphate buffer pH 8.0 and 5 mM β-mercaptoethanol with a 6 kDa cut-off membrane (Spectrum Laboratories). Next day, protein was loaded onto a POROS 20HS (Thermo Fischer Scientific) cation exchange column equilibrated with 50 mM sodium phosphate buffer pH 8.0 and 5 mM β-mercaptoethanol. Protein was eluted with a 0–100% linear gradient (15 CV) of 2 M KCl (Carl Roth). Fractions of interest were further buffer exchanged against 25 mM HEPES buffer pH 7.5, 75 mM KCl, 75 mM NaCl, and further concentrated using a concentrator (Vivaspin, cut-off 5 kDa). Protein concentration was measured using an ND-1000 UV/Vis spectrophotometer (Nanodrop Technologies) and purity was assessed with SDS–PAGE. Protein was aliquoted and stored at −80 °C.

**Purification of Hsp90**. Hsp90 was purified as previously described[66]. We over-produced N-terminally His$_6$-tagged human Hsp90β recombinantly in *E. coli* BL21 Rosetta 2 (Novagen). Cells were harvested, flash-frozen in liquid nitrogen and stored at −80 °C until further usage. Pellets were thawed in a water bath at 37 °C and resuspended in 12.5 mM sodium phosphate buffer pH 6.8 (Sigma-Aldrich), 75 mM KCl (Sigma-Aldrich), 1/2 tablet/50 ml EDTA-free protease inhibitor (Roche), 5 mM β-mercaptoethanol (Sigma-Aldrich). Cells were disrupted by an EmulsiFlex-C5 cell disruptor (Avestin). Lysate was cleared by centrifugation, filtered with a 0.22 μm polypropylene filter (VWR) and supernatant was purified using an ÄKTA purifier chromatography system (GE Healthcare). First, protein was loaded onto a POROS 20MC (Thermo Fischer Scientific) affinity purification column in 50 mM sodium phosphate buffer pH 8.0, 400 mM KCl, 5 mM β-mercaptoethanol eluted with a 0–100% linear gradient (5 CV) of 1 M imidazole. Peak was loaded onto a POROS 20HQ (Thermo Fischer Scientific) anion exchange column equilibrated with 25 mM sodium phosphate buffer pH 7.2 and 5 mM β-mercaptoethanol. Protein was eluted with a 0–100% linear gradient (15 CV) of 2 M KCl. Fractions of interest were then loaded onto a HiTrap heparin column high performance (GE Healthcare) column equilibrated with 25 mM sodium phosphate buffer pH 7.2 and 5 mM β-mercaptoethanol. Protein was eluted with a 0–100% linear gradient (15 CV) of 2 M KCl and peak was buffer exchanged against 25 mM sodium phosphate buffer pH 7.2, 150 mM KCl, 150 mM NaCl, and 5 mM β-mercaptoethanol. Protein concentration was measured using an ND-1000 UV/ Vis spectrophotometer (Nanodrop Technologies) and purity was assessed with SDS–PAGE. Protein was aliquoted and stored at −80 °C.

**Formation of Tau fibrils**. Monomeric Tau-RD\* was aggregated in aggregation buffer and heparin low molecular weight (Santa Cruz Biotech), concentrations depending on the experiment and Tau:heparin ratio always kept 4:1. Aggregation was performed at 37 °C, shaking at 180 rpm, and aliquots were flash frozen at time points indicated in the text. Aliquots were then thawed on ice for downstream applications.

**Preparation of density gradients**. Density gradients were prepared according to an established procedure[67]. Gradients were formed by dissolving 10% and 40% sucrose (Sigma-Aldrich), in 25 mM HEPES pH 7.5, 75 mM KCl, 75 mM NaCl. Gradients were set up in polyallomer centrifuge tubes (Beckmann) by filling them to half height with 40% sucrose and topping them up with an equal amount of 10% sucrose. Gradients were formed by tilting the tubes horizontally for 3 h at room temperature and then tilting them back to vertical position. Tubes were stored overnight at 4 °C and samples were loaded as described for each experiment.

**Sample preparation for density gradients**. 2.5 μl of Tau-RD\* aggregates (37 μM of monomers) at different aggregation stages (time of aggregation: 0, 1, 8, 24 h) were loaded onto density tubes and subjected to centrifugation, 2.5 h at 200,000×*g* (Fig. 1a).

2.5 μl of Tau-RD\* fibrils (20 μM of monomers, timepoint 24 h) reacted for 1 h at 37 °C with either 7.5 μl of aggregation buffer or 7.5 μl of aggregation buffer plus 0.5 μM mouse HA-Map7 M1-S227, either wt or R$_{12}$K. Samples were then entirely loaded onto density tubes and subjected to centrifugation, 2.5 h at 200,000×*g* (Fig. 4d).

10 μl of Tau-RD\* (20 μM of monomers), aggregated for 16 h in the plate reader Spectramax i3 (Molecular Devices) in the presence of increasing concentration of

Hsp90 (0, 1, 5, and 10 μM) in aggregation buffer, were loaded completely onto density tubes and subjected to centrifugation, 2.5 h at 200,000×g (Fig. 5b).

**Dot blot analysis of density gradients.** Dot blot was performed using a dot blot apparatus (BioRad) and nitrocellulose membrane 0.1 μM (Sigma-Aldrich) washed with PBS. Twelve fractions were manually collected for each tube into a 96-deep well (Thermo Fisher Scientific), and each dot-blot well was filled with 150 μl of fraction using a multi-pipette. Fractions were pulled through by applying vacuum after 10 min of incubation with the membrane at room temperature. Nitrocellulose membranes were blocked with PBS-blocking buffer (LI-COR) and incubated with primary antibody, either monoclonal anti-FLAG M2 (F3165, Sigma Aldrich, working dilution 1:1000) or anti-HA 12CA5 mouse hybridoma (produced in-house), at room temperature for 1 h. After three washes with PBS, secondary antibody Donkey anti-mouse IgG IR Dye 800 conjugated (610-732-002, Rockland, 1:5000) was added at room temperature for 45 min. After additional two washes with PBS-T and one final wash with PBS, detection was performed using Odyssey CLx (LI-COR). Imaging was performed via Image Studio Lite (LI-COR). Uncropped gels are presented in Supplementary Fig. 9.

**Transmission electron microscopy.** Specimens were prepared for transmission electron microscopy using a negative staining procedure. Briefly, a 5 μl drop of sample solution was adsorbed to a glow-discharged (twice for 20 s, on a Kensington carbon coater) pioloform-coated copper grid, washed five times on drops of deionized water, and stained with two drops of freshly prepared 2.0% uranyl acetate, for 1 and 5 min, respectively, and subsequently air dried. Samples were imaged at room temperature using a Tecnai T20 LaB$_6$ electron microscope operated at an acceleration voltage of 200 kV and equipped with a 4K by 4K FEI Eagle camera. Images were acquired at a defocus value of 1.5 μm. Magnification/pixel size on specimen level: Fig. 1a 62,000 times/0.178 nm pix$^{-1}$; Fig. 5c right panel 62,000 times, 0.178 nm pix$^{-1}$; Fig. 5c right panel 100,000 times, 0.110 nm pix$^{-1}$.

**Animals.** All experiments were approved by the DEC Dutch Animal Experiments Committee (Dier Experimenten Commissie), performed in line with institutional guidelines of Utrecht University and were conducted in agreement with Dutch law (Wet op de Dierproeven, 1996) and European regulations (Directive 2010/63/EU). Female pregnant Wister rats were obtained from Janvier Laboratories.

**Rat brain extracts preparation.** Rat brain extracts were obtained from female adult rats and homogenized in 10× volume/weight in tissue lysis buffer (50 mM Tris–HCl, 150 mM NaCl, 0.1% SDS, 0.2% NP-40, and protease inhibitors). Brain lysates were centrifuged at 16,000×g for 15 min at 4 °C, and the supernatant was used for the affinity purification-mass spectrometry experiments.

**Affinity purification-mass spectrometry on brain extracts.** Tau-RD* monomers, oligomers, or fibrils (37 μM monomeric concentration, with or without 37 μM Hsp90, depending on the experiment) were incubated for 1 h at 4 °C with FLAG beads (Sigma) previously blocked in chicken egg albumin (Sigma). Beads were then separated using a magnet (Dynal, Invitrogen) and washed three times with aggregation buffer to remove unbound Tau-RD* and excess of albumin. Beads conjugated with the Tau-RD* aggregated proteins were then incubated with brain extracts for 1 h at 4 °C. Beads were then washed in washing buffer (20 mM Tris–HCl, 150 mM KCl, 0.1% TritonX-100) for five times to remove aspecific neuronal proteins. For MS analysis, beads were then resuspended in 15 μl of 4× Laemmli sample buffer (Biorad), boiled at 99 °C for 10 min and supernatants were loaded on 4–12% Criterion XT Bis–Tris precast gel (Biorad). The gel was fixed with 40% methanol and 10% acetic acid and then stained for 1 h using colloidal coomassie dye G-250 (Gel Code Blue Stain, Thermo Scientific). Briefly, each lane from the gel was cut into three pieces and placed in 1.5 ml tubes. They were washed with 250 μl of water, followed by 15 min dehydration in acetonitrile. Proteins were reduced (10 mM DTT, 1 h at 56 °C), dehydrated and alkylated (55 mM iodoacetamide, 1 h in the dark). After two rounds of dehydration, trypsin was added to the samples (20 μl of 0.1 g l$^{-1}$ trypsin in 50 mM ammoniumbicarbonate) and incubated overnight at 37 °C. Peptides were extracted with acetonitrile, dried down and reconstituted in 10% formic acid prior to MS analysis.

**Mass spectrometry analysis.** All samples were analyzed on an Orbitrap Q-Exactive mass spectrometer plus or on an Orbitrap Q-Exactive mass spectrometer HF (Thermo Fisher Scientific) coupled to an Agilent 1290 Infinity LC (Agilent Technologies). Peptides were loaded onto a trap column (Reprosil pur C18, Dr. Maisch, 100 μm × 2 cm, 3 μm; constructed in-house) with solvent A (0.1% formic acid in water) at a maximum pressure of 800 bar and chromatographically separated over the analytical column (Poroshell 120 EC C18, Agilent Technologies, 100 μm x 50 cm, 2.7 μm) using 90 min linear gradient from 7% to 30% solvent B (0.1% formic acid in acetonitrile) at a flow rate of 150 nl min$^{-1}$. The mass spectrometers were used in a data-dependent mode, which automatically switched between MS and MS/MS. After a survey scan from 375 to 1600 m/z the 10 or 12 most abundant peptides were subjected to HCD fragmentation. MS spectra were acquired with a resolution > 30,000, whereas MS2 with a resolution > 17,500.

**MS acquisition parameters.** Replicate 1—Tau aggregation EXP:
Orbitrap Q-Exactive plus, top 10 peaks, dynamic exclusion = 12 s.
MS1 parameters: resolution = 35,000, AGC target = 3e6, max.IT = 50 ms.
MS2 parameters: resolution = 17,500, AGC target = 5e4, max.IT = 120 ms.
Replicate 2—Tau aggregation EXP:
Orbitrap Q-Exactive plus, top 10 peaks, dynamic exclusion = 12 s.
MS1 parameters: resolution = 35,000, AGC target = 3e6, max.IT = 50 ms.
MS2 parameters: resolution = 17,500, AGC target = 5e4, max.IT = 120 ms.
Replicate 3—Tau aggregation EXP:
Orbitrap Q-Exactive HF, top 12 peaks, dynamic exclusion = 12 s.
MS1 parameters: resolution = 60,000, AGC target = 3e6, max.IT = 20 ms.
MS2 parameters: resolution = 30,000, AGC target = 1e5, max.IT = 50 ms.
Replicate A—Hsp90 EXP:
Orbitrap Q-Exactive HF, top 12 peaks, dynamic exclusion = 12 s.
MS1 parameters: resolution = 60,000, AGC target = 3e6, max.IT = 20 ms.
MS2 parameters: resolution = 30,000, AGC target = 1e5, max.IT = 50 ms.
Replicate B—Hsp90 EXP:
Orbitrap Q-Exactive HF, top 12 peaks, dynamic exclusion = 12 s.
MS1 parameters: resolution = 60,000, AGC target = 3e6, max.IT = 20 ms.
MS2 parameters: resolution = 30,000, AGC target = 1e5, max.IT = 50 ms.
Replicate C—Hsp90 EXP:
Orbitrap Q-Exactive plus, top 10 peaks, dynamic exclusion = 12 s.
MS1 parameters: resolution = 35,000, AGC target = 3e6, max.IT = 50 ms.
MS2 parameters: resolution = 17,500, AGC target = 5e4, max.IT = 120 ms.

**Gene Ontology analysis.** Proteins were classified using the enrichment analysis tool provided by Gene Ontology Consortium (GO Consortium, 2017[28]).

**Circular dichroism.** HA-tagged (YPYDVPDYA) mouse Map7 truncations (M1-S227, both wild type and R$_{12}$K) were buffer exchanged against chloride-free phosphate buffer pH 7.5, 150 mM NaF (Sigma-Aldrich), to an estimated concentration of 0.05 mg ml$^{-1}$. Spectra were obtained on a circular dichroism detector (JASCO), range from 180 to 250 nm, cut off at high tension over 700 V.

**ThioflavinT aggregation assay.** Aggregation of Tau-RD* (20 μM stock) in aggregation buffer (total volume 100 μl) was stimulated by 5 μM heparin low molecular weight in the presence of 60 μM ThioflavinT (Sigma) in transparent, lidded Greiner 96-well plates (Sigma-Aldrich). To assess the impact of Hsp90, samples were supplied with 0, 1, 5, or 10 μM of Hsp90. Fluorescent spectra were recorded every 10 min for 16 h with a SpectraMax i3 (Molecular Devices).

**Dot blot profiling.** Depending on the experiment, a rectangle encompassing fractions of interest was drawn via ImageJ software. Profiles were then obtained and plotted.

**Disorder prediction and amino acidic bias.** Disordered residues in Tau interactome protein were predicted using MetaDisorder predictor[29]. We defined disorder percentage of each interactor as the amount of disordered amino acids divided by total amount of amino acids.

To calculate aminoacidic biases in disordered regions, disordered regions of each aggregation-specific interactor were considered as a single string of amino acids. The frequency of each amino acid in each disordered string was then calculated and compared to its frequency in the disordered regions of the whole proteome[68]. The ratio of calculated over predicted frequency was then computed and expressed as a log$_2$ function for the whole set of aggregation-specific interactors, sorted per amino acid, with residues ordered by their abundance in the human proteome.

Boxplot and graphs of Fig. 3 were created with SPSS.

Statistical significance of Fig. 3b and c was tested using the Wilcoxon paired non-parametric t-test on measured paired to predicted data for each protein.

**PSMs data analysis (Proteome Discoverer).** Raw data files were converted to *.mgf files using Proteome Discoverer 1.4 software (Thermo Fisher Scientific). Database search was performed using the Uniprot rat database and Mascot (version 2.5.1, Matrix Science, UK) as the search engine. Carbamidomethylation of cysteines was set as a fixed modification and oxidation of methionine was set as a variable modification. Trypsin was set as cleavage specificity, allowing a maximum of two missed cleavages. Data filtering was performed using a percolator[69], resulting in 1% false discovery rate (FDR). Additional filters were search engine rank 1 and mascot ion score > 20.

To infer protein abundancy of each individual protein co-purified with Tau-RD*, we relied on total numbers of PSMs; PSMs of each identified protein were then normalized on the PSMs of the purified Tau-RD* in each condition. Ratios were then calculated between normalized PSMs of the protein co-purified with Tau-RD* at different time point of aggregation and normalized PSMs of the same protein at the first time point (Tau aggregation experiments) or between normalized PSMs of samples with Hsp90 and their relative controls (Hsp90 experiment). Ratios were then transformed into log2. Crapome[70] was used to

analyze Tau-RD* interacting binding proteins in three biological replicas. Proteins co-precipitated with Tau-RD* at 0 h were compared with controls (FLAG-beads). Proteins with a Fold Change calculation > 3 were considered significant proteins sequestered by Tau-RD* monomers. This unbiased analysis for scoring AP-MS data, generated a selection of proteins characterized by higher binding affinity for monomeric species (Fig. 2c, Supplementary Fig. 2). Proteins co-precipitated with Tau-RD* at 4 h were compared with proteins co-precipitated with Tau-RD* at 0 h without heparin. Proteins with a Fold Change calculation > 2 were considered significant proteins sequestered by Tau-RD* aggregates. This unbiased analysis for scoring AP-MS data generated a selection of proteins characterized by higher binding affinity for fibrillary species (Fig. 2d, Supplementary Fig. 3). Same selected proteins are also shown in Fig. 6b to highlight how Tau-RD* interactomes change upon incubation with Hsp90. Data analysis was conducted using Perseus or R, hierarchical clustering was performed within Perseus using Euclidian distance. Figure 2b: Quantifications of proteins detected at $t = 0, 1, 8$ and 24 h in replica 3 are represented in the heat map. Figure 2c: Heat map shows an unbiased selection of proteins co-precipitating with Tau-RD* monomers without heparin ($t_{0-}$) compared to control pull downs (empty FLAG-beads). Only proteins with a Fold Change calculation > 3 (FC-B, Crapome; by averaging the spectral counts across the selected controls) were considered enriched in Tau-RD* monomers without heparin ($t_{0-}$) compared to controls in the three different biological replicas. Values shown in the heat map refer to time points $t_0$, $t_1$, $t_8$, and $t_{24}$ of replica 3. Supplementary Fig. 2: Interactors of Tau-RD* monomers selected in Fig. 2c are shown with their relative quantifications across all time points in three biological replicas. Figure 2d Heat map shows an unbiased selection of proteins specifically co-precipitating with Tau-RD* aggregates ($t_4$) compared to Tau-RD* monomers without heparin ($t_{0-}$). Only proteins with a Fold Change calculation > 2 (FC-B, Crapome; by averaging the spectral counts across the selected controls) were considered enriched in Tau-RD* aggregates ($t_4$) compared to Tau-RD* monomers in the three different biological replicas. Values shown in the heat map refer to time points $t_0$, $t_1$, $t_8$, and $t_{24}$ of replica 3. Supplementary Fig. 3: Full list of interactors of Tau-RD* oligomers and fibrils ($t_4$) shown in Fig. 2d is shown with their relative quantifications across all time points in three biological replicas. Figure 6b: Bar graph shows same selected proteins of Supplementary Fig. 3 (significant proteins interacting with Tau-RD* aggregates $t_4$) and their average quantifications in Tau-RD* ($t_4$) compared to Tau-RD* + Hsp90 ($t_4$) across three different biological replicas. Ratios of normalized PSMs of interactors in Tau-RD* + Hsp90/ normalized PSMs of interactors in Tau-RD*−Hsp90 are graphically represented and shown as percentages. Negative values indicate a decreased binding affinity in presence of Hsp90, while positive ones indicate a higher affinity. Quantification of selected interactors of Tau-RD* monomers without heparin $t_{0-}$ (Fbxo3, Poldip2, Ctbp1, Ctbp2) is also included as additional control. Most of the interactors of Tau-RD* aggregates reduce their binding affinity in presence of Hsp90, whereas Fbxo3, Poldip2, Ctbp1, and Ctbp2 do not.

The full datasets are available in Supplementary Data 1 and 3.

**Intensity-based data analysis (MaxQuant)**. Raw spectra from the time-course experiments and from the hsp90 experiments were analyzed using MaxQuant (version 1.6.6.0)[71]. Database search was performed using the Uniprot rat database. Carbamidomethyl of cysteines was set as a fixed modification and oxidation of methionine was set as a variable modification. Trypsin was set as a proteolytic enzyme, allowing a maximum of two missed cleavages. PSM FDR and protein FDR were set at 0.01. Reverse and potential contaminant hits were removed.

To infer protein abundancy of each individual protein co-purified with Tau-RD*, protein group intensities were used for the quantification. Intensities were then normalized on the Intensity of the purified Tau-RD* in each condition. Missing values were replaced with a constant value corresponding to the minimum intensity measured in each single replicate. Normalized intensities were then logarithmically transformed (log2). Differences were calculated from intensity at each time point against its intensity at the first time point (Tau aggregation experiments) or differences are calculated between Intensity in samples with Hsp90 with their relative controls (Hsp90 experiments). Supplementary Fig. 4: Interactors of Tau-RD* monomers selected in Supplementary Fig. 2 are shown with their relative intensity-based quantifications across all time points in three biological replicas. Supplementary Fig. 5: Full list of interactors of Tau-RD* oligomers and fibrils ($t_4$) shown in Supplementary Fig. 3 is shown with their relative intensity-based quantifications across all time points in three biological replicas. Supplementary Fig. 8: Bar graph shows selected proteins of Supplementary Fig. 5 (significant proteins interacting with Tau-RD* aggregates) and their average quantifications in Tau-RD* ($t_4$) compared to Tau-RD* + Hsp90 ($t_4$) across three different biological replicas. Differences between log2_normalized intensities of interactors in Tau-RD* + Hsp90 and log2_normalized intensities of interactors in Tau-RD*−Hsp90 are graphically represented. Negative values indicate a decreased binding affinity in presence of Hsp90, while positive ones indicate a higher affinity. Quantification of selected interactors of Tau-RD* monomers without heparin $t_{0-}$ (Fbxo3, Poldip2, Ctbp1, Ctbp2) is also included as additional control. Almost all the interactors of Tau-RD* aggregates reduce their binding affinity in presence of Hsp90 whereas Fbxo3, Poldip2, Ctbp1, and Ctbp2 do not.

The full datasets are available in Supplementary Data 2 and 4.

**Reporting summary**. Further information on research design is available in the Nature Research Reporting Summary linked to this article.

## Data availability

The mass spectrometry proteomics datasets generated during the current study are available in the PRIDE repository[72] with the accession number: PXD015432. Supplementary Data 1–4 contain full lists of protein quantifications for the AP-MS experiments described. Supplementary Table 1 contains the list of primer sequences used in this study and Supplementary Table 2 contains additional information on reagents and software used in this work. The source data underlying Figs. 2b–d, 3a–c, 4c, 5a, b, and 6a, b and Supplementary Figs. 2–8 are provided as a Source Data file. Other data are available from the corresponding author upon reasonable request.

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

## Acknowledgements
We are grateful to Ineke Braakman for continuous support. We thank Madelon Maurice for collaboration in the Initial Training Network "WntsApp" (No. 608180)], supported by Marie-Curie Actions of the 7th Framework program of the EU. S.G.D.R. was further supported by the Internationale Stichting Alzheimer Onderzoek (ISAO; project "Chaperoning Tau Aggregation"; No. 14542) and a ZonMW TOP grant ("Chaperoning Axonal Transport in neurodegenerative disease"; No. 91215084). This work was supported by the Netherlands Organization for Scientific Research (NWO) through a VIDI grant for M.A. (723.012.102) and Proteins@Work, a program of the National Roadmap Large-scale Research Facilities of the Netherlands (project number 184.032.201). This research was also (partially) funded by the Netherlands Organization for Scientific Research (NWO; 723.012.102 and 184.034.019) and the Top Sector Chemie (CHEMIE.PGT.2018.008 and CHEMIE.PGT.2019.008).

## Author contributions
S.G.D.R. and L.F. conceived the study; S.G.D.R., C.C.H., M.A., F.G.F. and L.F. planned experiments; L.F., R.S., K.K., G.v.d.K., R.K, L.S.v.B. and W.J.C.G. did experiments; L.F., R.S., K.K., G.v.d.K., R.K. and W.J.C.G. analyzed data; S.G.D.R. and L.F. wrote the manuscript, with contributions of R.S., K.K., G.v.d.K., R.K., L.S.v.B., W.J.C.G., F.G.F., M.A. and C.C.H.

## Competing interests
The authors declare no competing interests.
