## [Peer Review File · Nature Communications]

Reviewers' Comments:

Reviewer #1:

Remarks to the Author:

In this manuscript, Ferrari et al show multiple novel features of how pathological aggregation can systematically affect interactions made by tau protein. Aggregation of a fibrillation prone mutant of tau's microtubule binding repeat region changes the interaction profile progressively as oligomers mature from nanoscale aggregates to mature fibers, showing increased binding to proteins enriched in disordered sequences, especially RNA binding proteins and proteins enriched in arginine and methionine. By testing arginine to lysine substitution on a representative binding partner they demonstrate that interactions are affected directly by the arginine content of the disordered region, analogous to the literature observation that phase separation of disordered, arginine rich RNA binding proteins into biological condensates can be blocked by lysine substitution. They then show that even though the interaction is arginine dependent it can be modulated by the chaperone activity of hsp90. Overall, this work shows that pathological oligomerization of tau results in a gain of function with physiological implications, where widespread rewiring of tau's interaction profile presents a novel and interesting potential mechanism for the pathology of fibrillation.

However, while the claims made in the paper are broadly supported and of significant interest, the authors have not discussed them in the context of some previous literature that I strongly feel needs to be addressed. Specifically, they observe that the fibril-interactome contains phase-separating proteins, and that arginine content required for binding tau aggregates is also implicated in phase separation, but they do not reference recent work showing that tau itself is a phase separating protein.

For example, it has been shown that wild type full length tau can nucleate tubulin by forming a phase separated liquid condensate [PMID: 28877466], and that tau's ability to phase separate originates with the microtubule binding repeat region [PMID: 28819146]. In addition, multiple studies claim that fibrillation of wild type tau is, under conditions tested, consistently preceded by phase separation, even when fibrillation is induced by the addition of heparin. There also appears to be a fundamental relationship between fibrillation and phase separation because mutations [PMID: 29472250] and post-translational modifications [PMID: 29734651] that prevent fibrillation can also prevent phase separation. There is also evidence that analogous to some other phase separating proteins tau itself can coacervate with RNA (specifically tRNA) [PMID: 28683104].

This fact that propensity to form functional biological condensates often correlates with propensity to form pathological fibrils is a frequently observed, fundamentally important, and poorly understood issue. In their discussion the authors point out that the interactions which drive pathological aggregation and the interactions which drive physiological aggregation are flip sides of the same coin, but if liquid-liquid phase separation of tau is the physiologically relevant mode of aggregation then that statement directly applies to this issue, and should be made in context. With context provided I would strongly recommend publication.

Other Points:

1. The methods employed are well suited for probing interactions to stable aggregates, and I would agree that the results likely represent a valid measurement of aggregate behavior. However, there is a claim in the literature that tau phase separation typically precedes and can be required for the formation of stable fibrillar states of tau [PMID: 28819146] [PMID: 28877466].

As a basic control it would be useful to demonstrate that condensates are not present in this system, and that this interaction behavior is truly on the other side of the coin. This could be done by showing that tau condensates do not form in the specific conditions used to induce fibrillation for this construct, or, if liquid-liquid phase separation is observed, by showing that the condensate

is dispersed back into the monomer state during the protocol.

2. While the progressive changes in interaction behavior are clear, it's not clear what the mode of binding is. The paper discusses the possibility that these interactions are co-aggregates, not stoichiometric interaction partners per-se but aggregates that can trigger aggregation in another protein such that they form an aggregated complex. Given this possibility, I think it would be very informative to show what the tau fibrils look like in the presence of Map7 1M-227S truncations.

3. Since the bound fraction is enriched for RNA binding proteins I think it would be useful, but not essential, to know if they maintain their interactions with RNA during the pulldown. And, if RNA is being pulled down by tau fibrils, whether the presence of RNA in the lysate affects the interaction profile.

4. During the discussion of pi-pi contacts it needs to be pointed out that the ability to form stacking interactions is not the only difference between arginine and lysine. As one example their hydrogen bonding behavior is geometrically distinct, such that arginine is naturally suited to forming multiple hydrogen bonds, especially to phosphates and carboxyl groups, in a way that lysine is not.

5. Similarly, in describing the role of arginine in mediating these interactions, it's worth pointing out that it is not obvious what the mechanism of that interaction with fibrils actually is. There are very few pi-groups in the tau sequence for arginine to form pi-pi interactions with, especially given that the positive charges in the tubulin binding repeat region are biased almost entirely towards lysine, and it's not clear how fibrillation would affect the contact availability for the pi-containing groups that are in the sequence.

Putting the sequence of the recombinant tau-RD protein in Figure 4 would be useful for this discussion.

6. A fold change of 2x isn't necessarily significant when spectral counts are low. While this probably affects very few proteins in the sets shown the authors should still provide data on the counts used for normalization or, when there are multiple controls, should attempt significance testing using the standard deviation among controls.

7. The discussion claims, for the purpose of understanding aggregation in Alzheimer, that "the impact of phosphorylation on aggregation is yet to be understood. New structural insight proved that phosphorylated residues do not contribute significantly to assembly of mature fibrils (41)."

They should put this in the context of recent findings that hyperphosphorylation of tau induces phase separation and that this condensation step can still contribute to the assembly of fibrils [PMID: 29472250].

[PMID: 28877466]

Hernández-Vega, Amayra, et al. "Local nucleation of microtubule bundles through tubulin concentration into a condensed tau phase." *Cell reports* 20.10 (2017): 2304-2312.

[PMID: 28819146]

Ambadipudi, Susmitha, et al. "Liquid-liquid phase separation of the microtubule-binding repeats of the Alzheimer-related protein Tau." *Nature communications* 8.1 (2017): 275.

[PMID: 29472250]

Wegmann, Susanne, et al. "Tau protein liquid-liquid phase separation can initiate tau aggregation." *The EMBO journal* 37.7 (2018).

[PMID: 29734651]

Ferreon, Josephine, et al. "Acetylation disfavors tau phase separation." *International journal of molecular sciences* 19.5 (2018): 1360.

[PMID: 28683104]

Zhang, Xuemei, et al. "RNA stores tau reversibly in complex coacervates." *PLoS biology* 15.7 (2017): e2002183.

Reviewer #2:

Remarks to the Author:

The manuscript deals with a very important topic, and one which the authors correctly point out, has not been well studied. I believe it would be of interest to journal readers. However, some revisions would improve the manuscript and add further information.

1. In Figure 1 it would be helpful to include EM images from some of the other fractions to confirm the lack of longer fibrils and show the oligomeric species.

2. The authors should also discuss the choice of tau to a greater extent. The repeat domain is widely used, but in the context of these studies do the authors believe that the N and C terminals would not affect the results?

3. All of the abbreviations should be defined on first use.

4. In Figure 4 the authors should provide a quantification of the differences in the levels of Map7 in the different sucrose gradient fractions.

5. Similarly, in Figure 5 additional data would be helpful. Were the ThT reactions performed with replicates? If so the graphs should show the average plus error bars. Panel B should also be quantified.

6. In the discussion the authors should expand the potential effects of altered tau binding. COPI in particular is mentioned in the manuscript, how do the authors hypothesize that increased binding to oligomers would affect its function? Researchers have hypothesized that oligomers are the tau form which spreads between cells, are there binding properties that would make them more likely to be released or seed aggregation? Several protein kinases are bound by the mature filaments. In addition to the potential impact on tau phosphorylation, what other systems are possibly affected? How would these tie into the changes seen in AD and other tauopathies in early and late stage disease?

Reviewer #3:

Remarks to the Author:

Ferrari et al. present a study that aims to provide evidence of the importance of Arginine residues for the aggregation of proteins with Tau. This is a crucial step towards the understanding of protein aggregation processes for neurodegenerative diseases such as Alzheimer's. A key part of the manuscript is the quantitative proteomic analysis of protein-interactions during the aggregation process of Tau into Fibrils. Here the authors demonstrate time-dependent variations using quantitative mass spectrometry. Based on the data the authors conclude, among others, that Arginine residues might play a key role in this interaction process due to its overrepresentation in intrinsically disordered regions. With the creation of mutant proteins that have their relevant Arginine residues replaced by Lysine residues, the importance of the guanidium

group is clearly demonstrated. Finally yet importantly, HSP90 is shown as a clear modulator of the protein aggregation process with a significant impact on the interactome compared to the previous experiments. The presentation of the study is very logical and the thought process behind each experiment is very clear. However, I have some major comments in regard of the quantitative mass spectrometric part of the manuscript.

The authors applied the concept of "spectral counting" to retrieve label-free quantitative information. Even with the claim of this being an unbiased approach. However, I have some concerns in its usage in the presented study. One major drawback of counting PSMs is the over-estimation of high molecular weight proteins. Since, this proteins result in more peptides, in consequence, more PSMs will be counted in their favor. The opposite is true for smaller proteins. There are spectral counting algorithms in place that can correct for this effect, but it was not stated that one has been used in the present study. Further, the amount of spectra can largely be influenced by the complexity of the analyzed sample. Here the authors neglect completely to give the reader information regarding the overall amount of identified proteins per sample (e.g. in a supplemental table or figure). In consequence, a decrease in PSMs could be just due to the fact of having a higher complex sample. Referring to the data in Fig 6B: The loss for certain proteins could be solely attributed by the presence of HSP90 in the sample during the MS analysis. A protein resulting in >100 unique peptides (www.proteomicsDB.org). A normalization to the amount of Tau protein will not circumvent this problem. The authors do not state if or if not a cut-off for quantification was used. Were proteins that had only one unique peptide or PSM used for analysis? Overall, a label-free quantification approach using the area-under-the-curve of extracted ion chromatograms would be less prone to this errors and should be used at least in comparison to gain more trust into the presented data if the mentioned concerns can not be address otherwise.

The technical description of the MS experiment needs significant work, too. The authors state that three gel pieces (runtime gel?) per sample have been analyzed. The authors write either Top 10 or Top 20 peaks were used for MS/MS acquisition, but do not say for which experiment what set-up was applied. Mixing this settings is going to have a significant influence on the PSM values and thus on quantification. Only a few MS acquisition parameters were mentioned, e.g. the resolution for the respective scans. Here the authors write that the 15 000 resolution scan is the "sensitive" option. Which is not true. It is in fact the faster method and thus might be even less sensitive. Sensitivity here depends on a combination of settings e.g. on the maximum injection time, the amount of ions required (AGC target) and the resolution. The two other settings are not provided. Another - in context of PSM quantification - crucial setting is the dynamic exclusion, this is also missing.

The part "DATA AND SOFTWARE AVAILABILITY" should be revised completely. The presented content does not fit here and the data of the experiment are not available at e.g. PRIDE. Neither the raw data nor the table outputs from Proteome Discoverer.

Overall, I advise the authors to rework the proteomic data and the provided information. Since, in the current status reproducing this data is not possible.

Point-by-point reply

Ferrari et al.

Below we reproduce the reviewers' comments in full (blue, italic) and address them one-by-one (black)

Reviewer #1

In this manuscript, Ferrari et al show multiple novel features of how pathological aggregation can systematically affect interactions made by tau protein. Aggregation of a fibrillation prone mutant of tau's microtubule binding repeat region changes the interaction profile progressively as oligomers mature from nanoscale aggregates to mature fibers, showing increased binding to proteins enriched in disordered sequences, especially RNA binding proteins and proteins enriched in arginine and methionine. By testing arginine to lysine substitution on a representative binding partner they demonstrate that interactions are affected directly by the arginine content of the disordered region, analogous to the literature observation that phase separation of disordered, arginine rich RNA binding proteins into biological condensates can be blocked by lysine substitution. They then show that even though the interaction is arginine dependent it can be modulated by the chaperone activity of hsp90. Overall, this work shows that pathological oligomerization of tau results in a gain of function with physiological implications, where widespread rewiring of tau's interaction profile presents a novel and interesting potential mechanism for the pathology of fibrillation.

However, while the claims made in the paper are broadly supported and of significant interest, the authors have not discussed them in the context of some previous literature that I strongly feel needs to be addressed. Specifically, they observe that the fibril-interactome contains phase-separating proteins, and that arginine content required for binding tau aggregates is also implicated in phase separation, but they do not reference recent work showing that tau itself is a phase separating protein.

For example, it has been shown that wild type full length tau can nucleate tubulin by forming a phase separated liquid condensate [PMID: 28877466], and that tau's ability to phase separate originates with the microtubule binding repeat region [PMID: 28819146]. In addition, multiple studies claim that fibrillation of wild type tau is, under conditions tested, consistently preceded by phase separation, even when fibrillation is induced by the addition of heparin. There also appears to be a fundamental relationship between fibrillation and phase separation because mutations [PMID: 29472250] and post-translational modifications [PMID: 29734651] that prevent fibrillation can also prevent phase separation. There is also evidence that analogous to some other phase separating proteins tau itself can coacervate with RNA (specifically tRNA) [PMID: 28683104].

This fact that propensity to form functional biological condensates often correlates with propensity to form pathological fibrils is a frequently observed, fundamentally

important, and poorly understood issue. In their discussion the authors point out that the interactions which drive pathological aggregation and the interactions which drive physiological aggregation are flip sides of the same coin, but if liquid-liquid phase separation of tau is the physiologically relevant mode of aggregation then that statement directly applies to this issue, and should be made in context. With context provided I would strongly recommend publication.

We thank the reviewer for highlighting the importance of our study! It is an excellent suggestion to consider the link between phase separation and fibril formation in the Discussion in more depth. We are grateful for these suggestions! Thus, we followed the suggestion of this reviewer to expand the discussion with a **new paragraph on the phase separation properties of Tau itself**, including the literature brought up by this reviewer:

“Interestingly, Tau fibril-specific interactome contains proteins from the Heteronuclear Ribonucleoprotein family (**Fig. 2A** and **Supplementary Figure 3**), whose protein architecture promotes phase separation (Harrison et al., 2017). Thus, Tau fibrils can attract phase-separating proteins, potentially increasing their local concentration and promoting their co-aggregation. Notably, also Tau and Tau-RD can phase-separate before forming mature, downstream fibrils (Ambadipudi et al., 2017; Wegmann et al., 2018). Fibrillation and phase separation may be mechanistically related at molecular level as both mutations and post-translational modifications that prevent fibrillation can also prevent phase separation (Ferreon et al., 2018; Wegmann et al., 2018). Such Tau condensates may also induce aberrant condensation of partner proteins (Hernandez-Vega et al., 2017), which could be an intriguing factor modulating the interactome. However, a significant contribution of Tau droplets to the binding of phase-separating proteins observed in our study is unlikely because we generated aggregates at physiological ionic strength, which precludes phase-separation of Tau (Ambadipudi et al., 2017; Zhang et al., 2017). It did not escape our attention, though, that both phase separation and interactome modulation of fibrils both rely on π - π stacking. Taken together, our data mechanistically support a tight crosstalk between physiological and pathological aggregates. In terms of fundamental chemical contacts, physiological and pathological aggregates are two sides of the same coin.”

Other Points:

1. The methods employed are well suited for probing interactions to stable aggregates, and I would agree that the results likely represent a valid measurement of aggregate behavior. However, there is a claim in the literature that tau phase separation typically precedes and can be required for the formation of stable fibrillar states of tau [PMID: 28819146] [PMID: 28877466].

*As a basic control it would be useful to demonstrate that **condensates are not present in this system**, and that this interaction behavior is truly on the other side of the coin. This could be done by showing that tau condensates do not form in the specific conditions used to induce fibrillation for this construct, or, if liquid-liquid phase separation is observed, by showing that the condensate is dispersed back into the monomer state during the protocol.*

We thank the reviewer for this suggestion. In fact, we tried to stimulate LLPS and estimate the impact of phase separation in the context of our study, using the protocols of the Alberti group. It turned out that **Tau-RD did not form liquid droplets under the conditions we used in our experiments**. Two of the studies referred to by this reviewer showed that increasing ionic strength prevent LLPS formation of Tau, both for the full-length Tau (Zhang et al., 2017 [PMID: 28683104]) and Tau-RD (Ambadipudi et al., 2017 [PMID: 28819146]). For the present study, we

generated Tau-RD fibrils at physiological ionic strength (150 mM NaCl), when formation of Tau condensates is strongly impaired. Thus, consistent with the findings reported in both papers **we did not find any signs of LLPS at conditions used in this study**. This is now indicated in the discussion in the new paragraph on phase separation::

“However, a significant contribution of Tau droplets to the binding of phase-separating proteins observed in our study is unlikely because we generated aggregates at physiological ionic strength, which precludes phase-separation of Tau (Ambadipudi et al., 2017; Zhang et al., 2017). It did not escape our attention, though, that both phase separation and interactome modulation of fibrils both rely on π - π stacking.”

2. While the progressive changes in interaction behavior are clear, it's not clear what the mode of binding is. The paper discusses the possibility that these interactions are co-aggregates, not stoichiometric interaction partners per-se but aggregates that can trigger aggregation in another protein such that they form an aggregated complex. Given this possibility, I think it would be very informative to show what the tau fibrils look like in the presence of Map7 1M-227S truncations.

It appears that there is a misunderstanding that we now clarified in the revised manuscript. For the experiments analysing the interactions of Map7-M1-S227 reported in Fig. 4, we first generated the fibrils without Map7 1M-227S and next, in a separated experiment, we tested the binding between fibrils and Map7 1M-227S. Thus, our **setup excludes co-aggregation dynamics** and reveals instead stoichiometric interactions. We now adapted the text to avoid this misunderstanding: “Next, we set out to test whether the exchange of Arginines to Lysines would affect binding to fibrils. To this end, we first generated Tau-RD* fibrils and then tested their chemical binding to Map7 truncations.”

3. Since the bound fraction is enriched for RNA binding proteins I think it would be useful, but not essential, to know if they maintain their interactions with RNA during the pull-down. And, if RNA is being pulled down by tau fibrils, whether the RNA in the lysate affects the interaction profile.

We have investigated this but we concluded that it is very unlikely that RNA mediated interactions play a significant role for the pull downs. We indeed tried to purify and sequence RNA after pull downs but we failed to obtain sufficient quantities for sequencing. A crucial factor may be that we generated the neuronal lysates in the absence of RNase inhibitor. As RNA is very unstable, it **very likely that RNA was mostly degraded** even before lysates were mixed with fibrils. We agree that it would be interesting to address RNA contribution to fibril binding in the future, but it is beyond the scope of the present study.

4. During the discussion of pi-pi contacts it needs to be pointed out that the ability to form stacking interactions is not the only difference between arginine and lysine. As one example their hydrogen bonding behavior is geometrically distinct, such that arginine is naturally suited to forming multiple hydrogen bonds, especially to

phosphates and carboxyl groups, in a way that lysine is not.

Done. We clarified this in the text and by adding **new Fig. 4B** to avoid potential misunderstandings. The reviewer is correct that Arginine can coordinate phosphate and carboxyl groups more effectively than Lysine. However, it is unlikely that this activity plays a major role for the Tau interactome: (i) our Tau construct **does not have phosphate groups**, and notably (ii) the **Tau fragment we used is positively charged**, thus negatively charged residues are depleted, making it unlikely that recognition of carboxyl groups is a major factor modulating the interactome. We added Tau sequence as **new Fig. 4B** to make it visually clear to the reader. We also further clarified this point in the discussion:

“Arginine is better suited than Lysine to forming multiple hydrogen bonds, especially to phosphates and carboxyl groups. However, negatively charged side chains are depleted in the positively charged Tau-RD, excluding a major role for Coulomb interaction between tau fibrils and Arginine-rich interactors (**Fig. 4B**).”

5. Similarly, in describing the role of arginine in mediating these interactions, it's worth pointing out that it is not obvious what the mechanism of that interaction with fibrils actually is. There are very few pi-groups in the tau sequence for arginine to form pi-pi interactions with, especially given that the positive charges in the tubulin binding repeat region are biased almost entirely towards lysine, and it's not clear how fibrillation would affect the contact availability for the pi-containing groups that are in the sequence.

Putting the sequence of the recombinant tau-RD protein in Figure 4 would be useful for this discussion.

Done. We thank the reviewer for bringing up this point, which we addressed in the **new Fig. 4B** and in the adapted Discussion. Tau-RD has a **specific motif that is known to engage in pi-stacking**, namely Val-Pro-Gly-Xxx-Gly, which we highlight in the new Fig. 4B, together with other potential pi-stacking positions (Vernon et al., 2018; Yeo et al., 2011). Fibrillization stacks these motives, potentially increasing the avidity of a binder to fibrils compared to monomers. We edited the text in the discussion accordingly to make this point explicit:

“We mapped potential π -stacking residues in the Tau sequence, revealing 10 potential sites, including a VPGGG motif, which is known to engage in π -stacking in other context (Vernon et al., 2018; Yeo et al., 2011) (**Fig. 4B**). Tau fibrillization stacks these sites, possibly increasing the avidity of Arginine-rich binders and favouring their binding to fibrils over monomers. Thus, our work describes interactome re-arrangements as a key molecular component driving interactome re-arrangements, namely Arginine side chains capable of engaging in π - π interactions”

6. A fold change of 2x isn't necessarily significant when spectral counts are low. While this probably affects very few proteins in the sets shown the authors should still provide data on the counts used for normalization or, when there are multiple controls, should attempt significance testing using the standard deviation among controls.

Done. We followed the suggestion of the reviewer and added **new Supplementary Tables S1-S4 containing data of the counts**, including the area-under-the-curve quantification in addition to PSM counting. Both methods reveal almost identical results.

7. The discussion claims, for the purpose of understanding aggregation in Alzheimer, that “the impact of phosphorylation on aggregation is yet to be understood. New structural insight proved that phosphorylated residues do not contribute significantly to assembly of mature fibrils (41).”

They should put this in the context of recent findings that hyperphosphorylation of tau induces phase separation and that this condensation step can still contribute to the assembly of fibrils [PMID: 29472250].

Done. We now cover this paper by Wegmann et al. in the Discussion, and we **revised the paragraph** as follows:

“The fact that Tau fibrils attract kinases and other phosphorylation regulators relates to hyperphosphorylation of Tau in Alzheimer (Wang & Mandelkow, 2016; Goedert et al. 2017). Our data show that Tau fibrils target both kinases and protein phosphatases, two classes of proteins crucial for neuronal survival during neurodegeneration (Miloso et al., 2008; Das et al., 2015). Interaction of Tau fibrils with these proteins may contribute to disrupt pro-survival pathways. While phosphorylation speeds up fibril formation of Tau, remarkably Tau fibrils isolated from patients’ brain do not show regular phosphorylation patterns that would be visible in the structure (Fitzpatrick et al., 2017; Tepper et al., 2014). Conversely, recent findings revealed how Tau hyperphosphorylation induces phase separation and consequent assembly of Tau fibrils (Wegmann et al., 2018). However, the mechanistic impact of phosphorylation on Tau aggregation is yet to be understood.”

Reviewer #2

*The manuscript deals with a **very important topic**, and one which the authors correctly point out, has not been well studied. I believe it would be **of interest to journal readers**. However, some revisions would improve the manuscript and add further information.*

Thank you for your kind assessment!

1. In Figure 1 it would be helpful to include EM images from some of the other fractions to confirm the lack of longer fibrils and show the oligomeric species.

Done. **New supplementary Figure S1** shows additional images of aggregating **Tau-RD* at 1.5 and 4.5 h**. Note that at the earliest timepoint we did not observe fibrils. S1).

2. The authors should also discuss the choice of tau to a greater extent. The repeat domain is widely used, but in the context of these studies do the authors believe that the N and C terminals would not affect the results?

Done. We now elaborate in more detail about the **choice of the fragment** and its consequences in a **new paragraph in the Discussion**:

“We focused on the interactions established by Tau-RD, as this stretch harbours the amyloid toxic fold (Zhang et al., 2019; Fitzpatrick et al., 2017). Therefore, the interactome we obtained gives a comprehensive picture of amyloid-associated binders. our approach may have miss potential interactors of the less hydrophobic N- and C-terminal stretches missing in Tau-RD. However, as aggregation of the Tau-RD fragment causes memory loss *in vivo* (Mocanu et al., 2008), it is likely that interactors of the amyloidogenic core region are potentially most relevant for neurodegeneration.”

3. All of the abbreviations should be defined on first use.

Done. We carefully **checked all abbreviations** and defined them on first use.

4. In Figure 4 the authors should provide a quantification of the differences in the levels of Map7 in the different sucrose gradient fractions.

Done. We **provided quantification of Map7** in density gradients as **new Supplementary Figure S6**.

5. Similarly, in Figure 5 additional data would be helpful. Were the ThT reactions performed with replicates? If so the graphs should show the average plus error bars. Panel B should also be quantified.

Done. As suggested, we **provided quantification of the fibril fractions** in Fig. 5 as new inset within this figure. We also added **new Supplementary Figure S7** showing data for an **entirely independent aggregation experiment** using different protein batches both for Tau and Hsp90, to back up the data in Fig. 5. Both independent experiments show that Hsp90 decreases tau aggregation rates. Please note that variations between aggregation experiments are inherent as they depend on stochastic early-aggregation events. That is why we prefer to show the experiments separately instead of averaging the two aggregation events.

*6. In the discussion the authors should **expand the potential effects of altered tau binding**. COPI in particular is mentioned in the manuscript, how do the authors hypothesize that increased binding to oligomers would affect its function? Researchers have hypothesized that oligomers are the tau form which spreads between cells, are there binding properties that would make them more likely to be released or seed aggregation? Several protein kinases are bound by the mature filaments. In addition to the potential impact on tau phosphorylation, what other systems are possibly affected? How would these tie into the changes seen in AD and other tauopathies in early and late stage disease?*

Done. We followed the suggestion of the reviewer and **expanded the discussion** on the potential **impact of altered Tau interactions** inside cells:

“We show that early-stage aggregates specifically attract COPI components. These interactors decrease in abundance as fibrils form, highlighting the exchange of interactors as aggregation proceeds and the specific reactivity of Tau oligomers. Our data can explain Golgi fragmentation observed in neurons of Alzheimer brains without tangles (Stieber et al., 1996). They also suggest a route for Tau oligomers to interact with the vesicular transport system, which may support release outside neurons and some prion-like activities (Clavaguera et al., 2017).”

And:

“The fact that Tau fibrils attract kinases and other phosphorylation regulators relates to hyperphosphorylation of Tau in Alzheimer (Wang&Mandelkow, 2016; Goedert et al., 2017). Our data show that Tau fibrils target both kinases and protein phosphatases, two classes of proteins crucial for neuronal survival during neurodegeneration (Miloso et al., 2008; Das et al., 2015). Interaction of Tau fibrils with these proteins may contribute to disrupt pro-survival pathways.”

Reviewer #3

Ferrari et al. present a study that aims to provide evidence of the importance of Arginine residues for the aggregation of proteins with Tau. This is a crucial step towards the understanding of protein aggregation processes for neurodegenerative diseases such as Alzheimer's. A key part of the manuscript is the quantitative proteomic analysis of protein-interactions during the aggregation process of Tau into Fibrils. Here the authors demonstrate time-dependent variations using quantitative mass spectrometry. Based on the data the authors conclude, among others, that Arginine residues might play a key role in this interaction process due to its overrepresentation in intrinsically disordered regions. With the creation of mutant proteins that have their relevant Arginine residues replaced by Lysine residues, the importance of the guanidinium group is clearly demonstrated. Finally yet importantly, HSP90 is shown as a clear modulator of the protein aggregation process with a significant impact on the interactome compared to the previous experiments. The presentation of the study is very logical and the thought process behind each experiment is very clear. However, I have some major comments in regard of the quantitative mass spectrometric part of the manuscript.

We very much appreciate that the reviewer is so positive on the significance of our work and the presentation of the study! This reviewer also raises some concerns on the mass spectrometric data, which we address in full in the revised version, as outlined below in detail.

The authors applied the concept of “spectral counting” to retrieve label-free quantitative information. Even with the claim of this being an unbiased approach. However, I have some concerns in its usage in the presented study. One major drawback of counting PSMs is the over-estimation of high molecular weight proteins. Since, this proteins result in more peptides, in consequence, more PSMs will be counted in their favor. The opposite is true for smaller proteins. There are spectral

counting algorithms in place that can correct for this effect, but it was not stated that one has been used in the present study.

Done, we added **new supplementary figures 4, 5 and 8** to address these concerns. We thank the reviewer for pointing this out. Overestimation of high-molecular weight proteins is actually less of a problem when using spectral counting for the relative quantification of the same protein between different samples, as it is done here. Nevertheless, to exclude any bias caused by the quantification based on PSMs, we **repeated all MS-based quantifications** by re-performing the data search using Max Quant and consequently by applying **intensity based-approaches** for the data analyses. In the current version of the manuscript we have added new supplementary Figures (Supplementary Figures 4-5, and 8) to support our previous quantification results of Tau interactome (Supplementary Figures 4-5) and of Hsp90 effect of Tau interactome (Supplementary Figure 8). The measurements of protein abundances obtained by applying intensity-based methods are **practically identical to previous results in which we used spectral counting (PSMs)** as quantification method (Supplementary Figures 2-3, Figure 6). Based on these considerations we could conclude that our quantifications are very robust and are not affected by any drawback caused by the type of quantification used for the data analysis.

Further, the amount of spectra can largely be influenced by the complexity of the analyzed sample. Here the authors neglect completely to give the reader information regarding the overall amount of identified proteins per sample (e.g. in a supplemental table or figure).

Done. We fully agree with the reviewer that this information should be in a supplementary table. We are sorry that the Tables with all the details of the quantifications were not accessible after the submission. Due to a technical problem they were not linked to the first version of the manuscript. We have now linked the “old” quantification results based on PSMs (**Supplementary Tables S1 and S2**) alongside with the new results of the analyses based on label-free quantifications on intensities (**Supplementary Tables S3 and S4**).

In consequence, a decrease in PSMs could be just due to the fact of having a higher complex sample. Referring to the data in Fig 6B: The loss for certain proteins could be solely attributed by the presence of HSP90 in the sample during the MS analysis. A protein resulting in >100 unique peptides (www.proteomicsDB.org). A normalization to the amount of Tau protein will not circumvent this problem.

Done. We now present **quantification results in the new Supplementary Tables S3 and S4 and new Supplementary Figure S8** that show that the concern raised by the reviewer does not apply to the data set in our study. The data presented in Supplementary Table S3 demonstrate that the presence of Hsp90 in the sample cannot solely explain the decrease in PSMs measured for specific interactors of fibrillary Tau species. In fact **only a restricted group of proteins** (which for the most part are interactors increasing in abundance upon Tau aggregation) **showed a**

decreased abundance in *in vitro* assays performed in presence of Hsp90, whereas other proteins are not affected (Supplementary Table S3). To further highlight this aspect, a **selected group of proteins interacting with Tau monomers** (Fbxo3, Ctb2, Ctb1 and Poldip2), which should not be influenced by the effect of Hsp90 on Tau conformation, have been included in the bar graph in Figure 6B. As expected, **their abundances do not change in the presence/absence of the chaperone, thereby confirming the specificity of the Hsp90 mediated effect on the fibrillary Tau interactome**. To overcome potential drawbacks caused by PSM quantification, a similar bar graph has been generated using an intensity-based quantification method (Supplementary Figure S8, Supplementary Table S4) showing almost identical results. Only the fibrillary Tau interactome is re-shaped in presence of Hsp90 while control proteins (Fbxo3, Ctb2, Ctb1 and Poldip2) do not show any significant change.

The authors do not state if or if not a cut-off for quantification was used. Were proteins that had only one unique peptide or PSM used for analysis?

For the PSMs analysis shown in Figure 6B we did not include an extra cut-off since **all the proteins shown here were previously selected as “true interactors”** of Tau-RD* aggregates in the previous analysis. Crapome has been used to calculate a Fold Change enrichment (FC) compared to the control Tau-RD* monomers (Tau-T0) (**Supplementary Figure S4, Supplementary Table S1**).

Overall, a label-free quantification approach using the area-under-the-curve of extracted ion chromatograms would be less prone to this errors and should be used at least in comparison to gain more trust into the presented data if the mentioned concerns can not be address otherwise.

Done. In parallel to PSM quantifications, **we indeed performed complementary label free quantifications** to overcome possible bias due to PSM counting. We now present the full results of the new analysis calculated on intensities in the **new Supplementary Tables S2 and S4**. We conclude that quantification results from **both methods (PSMs-based and intensity-based) are almost identical**, further strengthening the robustness of the *in vitro* AP-MS experiments performed.

The technical description of the MS experiment needs significant work, too. The authors state that three gel pieces (runtime gel?) per sample have been analyzed. The authors write either Top 10 or Top 20 peaks were used for MS/MS acquisition, but do not say for which experiment what set-up was applied. Mixing this settings is going to have a significant influence on the PSM values and thus on quantification. Only a few MS acquisition parameters were mentioned, e.g. the resolution for the respective scans. Here the authors write that the 15 000 resolution scan is the “sensitive” option. Which is not true. It is in fact the faster method and thus might be even less sensitive. Sensitivity here depends on a combination of settings e.g. on the maximum injection time, the amount of ions required (AGC target) and the

resolution. The two other settings are not provided. Another - in context of PSM quantification - crucial setting is the dynamic exclusion, this is also missing.

Done. We thank the reviewer for these comments. We **revisited and corrected the methodology** part following the inputs and suggestions given (the revised MS section in the Methods now extends to more than three manuscript pages (p. 17 and pp. 19-21)).

The part "DATA AND SOFTWARE AVAILABILITY" should be revised completely. The presented content does not fit here and the data of the experiment are not available at e.g. PRIDE. Neither the raw data nor the table outputs from Proteome Discoverer.

Done. We followed the suggestions of the reviewer and **made available raw data, search data and table outputs** generated with Proteome Discoverer (PSMs analysis) or MaxQuant (Intensity analysis) **on PRIDE**. So for the reviewer it is now possible to visualize the details of each single MS experiments by logging in using **these credentials**:

Pride accession number: PXD015432

Username: reviewer83099@ebi.ac.uk

Password: 9TNg59D8

Overall, I advise the authors to rework the proteomic data and the provided information. Since, in the current status reproducing this data is not possible.

Done. The proteomic data have been **re-structured following reviewer's advice**. **Extra data analyses** have been performed using MaxQuant and label free quantification. **New tables** with MS data are now linked to the manuscript and all the MS results are available on the **PRIDE repository**.

Reviewers' Comments:

Reviewer #1:

Remarks to the Author:

The authors' response more than addresses my concerns.

The rewiring of tau's interactions, and the mechanistic and direct relationships to phase separation and protein chaperone systems, are novel and important findings in of themselves, but this study also provides a significant resource in terms of labeling proteins and pathways whose function may be pathologically affected by co-aggregation with tau fibrils.

Reviewer #3:

Remarks to the Author:

The authors made a thorough follow up on the previous raised concerns and significantly improved the MS part of the manuscript. I am very pleased to see that the authors results being consistent regardless of the analysis method chosen. This should only strengthen and underscore the potent results presented here.